# From Absolute to Relative: Rethinking Reward Shaping in Group-Based Reinforcement Learning

Wenzhe Niu [*†12] Wei He [*2] Zongxia Xie [1] Jinpeng Ou [2] Huichuan Fan [2] Yuchen Ge [2] Yanru Sun [1]
Ziyin Wang [1] Yizhao Sun [2] Chengshun Shi [2] Jiuchong Gao [2] Jinghua Hao [2] Renqing He [2]

## Abstract

Reinforcement learning has become a cornerstone for enhancing the reasoning capabilities of Large Language Models, where group-based approaches such as GRPO have emerged as efficient paradigms that optimize policies by leveraging intra-group performance differences. However, these methods typically rely on absolute numerical rewards, introducing intrinsic limitations. In verifiable tasks, identical group evaluations often result in sparse supervision, while in open-ended scenarios, the score range instability of reward models undermines advantage estimation based on group means. To address these limitations, we propose **Reinforcement Learning with Relative Rewards (RLRR)**, a framework that shifts reward shaping from absolute scoring to relative ranking. Complementing this framework, we introduce the **Ranking Reward Model**, a listwise preference model tailored for group-based optimization to directly generate relative rankings. By transforming raw evaluations into robust relative signals, RLRR effectively mitigates signal sparsity and reward instability. Experimental results demonstrate that RLRR yields consistent performance improvements over standard group-based baselines across reasoning benchmarks and open-ended generation tasks.

## 1. Introduction

In recent years, reinforcement learning (RL) has driven significant advances in multiple domains exemplified by natural language processing (NLP) (Chen et al., 2026a; Zhou et al., 2025; Niu et al., 2025; Jin et al., 2025), reshaping the reasoning paradigms of large language models (LLMs) (Wang et al., 2026; Chen et al., 2026c). Through large-scale RL training, models such as DeepSeek-R1 (Guo et al., 2025) and OpenAI O1 (Jaech et al., 2024) have demonstrated sophisticated reasoning abilities, which substantially improve their performance on challenging mathematical and programming tasks (Chen et al., 2026b; He et al., 2026). Building on this progress, Group Relative Policy Optimization (GRPO) (Shao et al., 2024) has emerged as a key method for scaling LLMs during testing. By introducing an intra-group relative evaluation mechanism, GRPO reduces the bias of value function estimation and alleviates the heavy memory requirements associated with traditional Proximal Policy Optimization (PPO) (Schulman et al., 2017), providing a more efficient and robust training paradigm for the next generation of LLMs.

Reward modeling constitutes a central component in Reinforcement Learning, yet the reliance on absolute signal values creates a structural bottleneck for group-based optimization methods such as GRPO. Since these algorithms depend fundamentally on within-group difference to estimate advantages, sparse scoring mechanisms often fail to provide stable and continuous learning signals. This mechanism implies that a group of responses contributes to learning only when the within-group reward variance is non-zero; otherwise, all advantages reduce to zero, yielding no policy gradient updates. Under rule-based verifiers that provide binary correctness feedback, all correct responses receive an identical reward of one and all incorrect responses receive zero. As the policy improves during training, an increasing proportion of prompt groups reach a state of unanimous correctness, where all $G$ sampled responses are correct and thus all receive the same reward. We designate such unanimously scored groups as *ineffective samples*, since their zero reward variance produces uniformly zero advantages. As illustrated in Figure 1a, the fraction of *effective samples* that yield mixed outcomes declines to below 40% in later training stages, implying that the majority of computational cost is expended on groups that contribute no policy gradient.

While employing Scalar Reward Models (SRMs) can mitigate this sparsity by providing dense feedback, it introduces

---

*Equal contribution †Work done during internship at Meituan. [1]Tianjin University, Tianjin, China [2]Meituan, Beijing, China. Correspondence to: Zongxia Xie <caddiexie@hotmail.com>.

*Proceedings of the 43rd International Conference on Machine Learning*, Seoul, South Korea. PMLR 306, 2026. Copyright 2026 by the author(s).

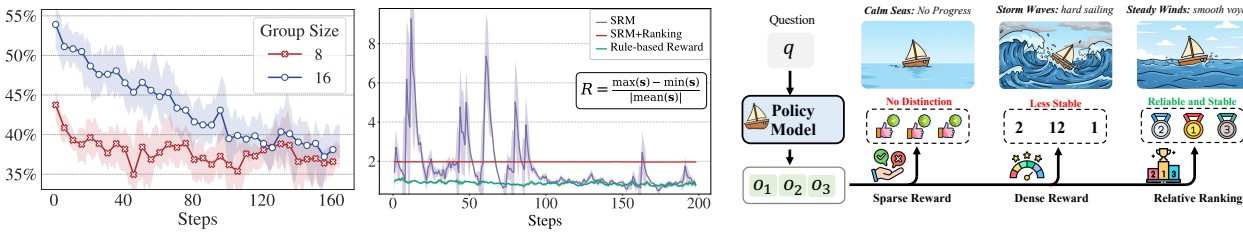

(a) Ratio of Effective Prompts      (b) Relative Range Dynamics      (c) Comparison of Different Rewards

*Figure 1.* Comparative analysis of reward formulations. (a) Sparse rewards cause zero intra-group variance, reducing data utilization efficiency during GRPO training. (b) Scalar reward models exhibit higher numerical dispersion, compromising stability in intra-group advantage estimation. (c) RLRR utilizes relative ranking within each group to generate fine-grained reward signals, avoiding both the zero-variance problem of sparse rewards and the numerical instability of scalar reward models, thereby sustaining stable advantages.

a distinct instability arising from the unbounded nature of scalar scores. The standard Bradley-Terry training objective used by SRMs exerts a constant gradient pressure to enlarge the reward gap between preferred and rejected responses. In the absence of explicit regularization on reward magnitudes, this pressure drives scores to diverge over the course of training. When such drifting scores are used in GRPO's group-normalized advantage estimation, a single outlier value can shift the group mean substantially, distorting the computed advantages for all other samples in the group. As shown in Figure 1b, the relative range of SRM scores exhibits significant instability throughout training, compounding this effect. Consequently, current absolute reward paradigms either suffer from vanishing gradients due to outcome homogeneity or unstable advantage estimation due to score sensitivity. We provide a detailed analysis of the impact of the SRMs' unboundedness on group-based reinforcement learning in Appendix B.

To address these limitations, we propose **Reinforcement Learning with Relative Rewards (RLRR)**, a framework designed to transition the reward shaping paradigm from absolute valuation toward relative ordering. By integrating intra-group ranking information directly into the advantage computation, RLRR effectively mitigates the dependency on the absolute magnitude of reward signals. We devise distinct relative reward mechanisms tailored to the availability of ground truth: for tasks equipped with rule-based verifiers, RLRR synergizes ranking signals with rule-based feedback; conversely, for tasks lacking deterministic verification, the framework relies exclusively on relative ranking. To facilitate this approach, we introduce the **Ranking Reward Model (Ranking RM)**, which is trained to discriminate the relative quality of multiple input samples. This design aligns with the inherent group-based comparison mechanism of GRPO. By employing the Ranking RM within the RLRR framework, our method is able to extract valid learning signals from sample groups that yield homogeneous outcomes under absolute scoring. This capability allows the model to continue learning from conventionally ineffective samples,

thereby significantly enhancing reasoning quality and maximizing the utilization of rollout data. We rigorously validate the effectiveness of our methodology through extensive experiments across diverse settings, demonstrating substantial improvements over existing approaches. Our contributions are summarized as follows:

- We propose RLRR, a framework that integrates relative rewards into group-based optimization to resolve signal sparsity in verifiable tasks and score instability in open-ended generation.

- We introduce the Ranking Reward Model, a listwise preference model that generates direct relative rankings, offering robust guidance immune to absolute score fluctuations.

- Extensive experiments validate the superiority of the relative reward paradigm, demonstrating consistent performance gains over absolute-scoring baselines across diverse benchmarks. Our source code is available at https://github.com/niuwz/RLRR

## 2. Related Work

**Inference-Time Scaling for LLMs.** Inference-time scaling complements training efforts, with research focusing on sampling and reward model aggregation (Brown et al., 2024; Snell et al., 2025; Wu et al., 2024). A key approach is Reinforcement Learning with Verifiable Reward (RLVR), which improves reasoning by using external verifiers for reward signals instead of model-generated scores (Zeng et al., 2025). Methods like PPO (Schulman et al., 2017) and GRPO (Shao et al., 2024) are commonly used for policy optimization, driving further RL advancements in reasoning tasks (Kazemnejad et al., 2024; Yuan et al., 2025). Notable innovations include DAPO, which filters zero-variance prompts (Yu et al., 2025), and GRESO, which uses probabilistic pre-filtering (Zheng et al., 2025). While both improve data efficiency, DAPO incurs computational overhead, and GRESO may discard useful learning opportunities due

to its simplistic reward structure. While some concurrent methods also leverage relative reward signals (Wang et al., 2025; Yang et al., 2025b), RLRR specifically targets sparse rewards and unstable reward ranges in group-based RL for language reasoning.

**Reward Models.** Reward models (RMs) are pivotal in RL, especially for aligning LLMs and scaling inference. Designed to capture human preferences, RMs complement rule-based rewards (Christiano et al., 2017; Ouyang et al., 2022). Mainstream RMs typically function as discriminative classifiers, providing scalar rewards to rank responses (Cai et al., 2024; Liu et al., 2025a; Lou et al., 2024). Other methods harness LLMs as judges, offering preference scores or critiques on generated content (Zheng et al., 2023). Approaches like Direct Preference Optimization (DPO) eliminate the need for explicit RMs, instead directly optimizing policies from preference pairs (Rafailov et al., 2023). Despite their advantages, RMs face challenges, such as the high cost of preference data, biases, and the risk of reward hacking (Gao et al., 2023; Skalse et al., 2022).

## 3. Preliminaries

To optimize the LLM policy, GRPO (Shao et al., 2024) introduces an alternative RL algorithm, which is a memory-efficient variant of PPO (Schulman et al., 2017). A notable feature of GRPO is that it typically operates without a learned value function. Instead, for a given prompt $p$, the current policy generates a group of $G$ responses $\{o_1, \ldots, o_G\}$. The rewards $\{s_1, \ldots, s_G\}$ for these responses are then used to compute the relative advantage for each response:

$$\hat{A}_k = \frac{s_k - \text{mean}(\{s_k | k = 1, 2, \ldots, G\})}{F_{\text{norm}}}. \quad (1)$$

Here, $F_{\text{norm}}$ serves as an optional normalization factor. In the standard GRPO implementation, $F_{\text{norm}}$ is defined as $\text{std}(\{s_k | k = 1, \ldots, G\})$. In contrast, alternative implementations in RLVR fix the normalization factor to unity so that $F_{\text{norm}} = 1$ (Liu et al., 2025b; Chu et al., 2025). This mean-based estimator is inherently sensitive to reward range, as a single outlier can shift the group mean and distort the advantages for all other samples.

GRPO then maximizes a clipped surrogate objective function to ensure stable updates. Let $\pi_{\theta_{\text{old}}}$ represent the policy before the update. For each token $o_{k,t}$ in a trajectory $o_k$ (from state $s_t$), the importance sampling ratio is defined as $\rho_{k,t}(\theta) = \frac{\pi_\theta(o_{k,t}|s_t)}{\pi_{\theta_{\text{old}}}(o_{k,t}|s_t)}$. The objective is then given by:

$$\mathcal{J}_{\text{GRPO}}(\theta) = \frac{1}{G} \sum_{k=1}^{G} \frac{1}{|o_k|} \sum_{t=1}^{|o_k|} \min\Big( \rho_{k,t}(\theta) \cdot \hat{A}_k,$$
$$\text{clip}\big(\rho_{k,t}(\theta), 1 - \epsilon, 1 + \epsilon\big) \cdot \hat{A}_k \Big), \quad (2)$$

where $\epsilon$ is a small hyperparameter defining the clipping range. This mechanism ensures that the LLM policy is updated while maintaining stable gradient constraints.

## 4. Methodology

To address the instability of absolute scoring in group-based optimization, we propose **Reinforcement Learning with Relative Rewards (RLRR)**. This framework anchors advantage estimation in intra-group rankings, providing robust relative signals that align with the comparative nature of the learning objective.

### 4.1. Reinforcement Learning with Relative Rewards

We introduce RLRR, a framework that integrates intra-group relative quality rankings into group-based reinforcement learning, such as GRPO. RLRR aims to mitigate gradient vanishing when reward variance collapses within the sampled response set $\{o_i\}_{i=1}^{G}$. Formally, given the group of responses, we assign each response a rank $r_i \in \{1, \ldots, r_{\max}\}$ based on its relative quality, where $r_i = 1$ denotes the best response and $r_{\max}$ represents the maximum rank index. We then synthesize these ranks into the final reward via a reward-shaping function $f(\cdot)$ for advantage computation. Depending on the availability of ground truth verification, we devise specific strategies to incorporate these relative rankings as either a fine-grained supplement to rule-based feedback or as the primary reward signal.

**Hybrid Relative Reward (HRR).** Tailored for tasks with verifiable outcomes (e.g., mathematical reasoning), HRR preserves the authoritative ground truth signal while introducing fine-grained preference information to resolve tie-breaking scenarios. We define the hybrid reward as a bounded correction to the binary rule-based score $s_i^{\text{rule}}$:

$$s_i^{\text{rank}} = f_{\text{HRR}}\big(s_i^{\text{rule}}, r_i\big) = s_i^{\text{rule}} + \tau \cdot \tanh\left(\frac{r_{\max}}{r_i} - 1\right), \quad (3)$$

where $\tau$ controls the magnitude of the rank-based adjustment. The hyperbolic tangent function provides a non-linear incentive that significantly boosts top-ranked responses while naturally limiting the correction range. This design ensures that the relative signal effectively differentiates samples sharing identical correctness labels without overriding the primary ground truth objective.

**Pure Relative Reward (PRR).** In tasks lacking reliable ground truth, absolute scores from reward models frequently exhibit high variance and range instability. PRR addresses this by replacing raw scalar evaluations with a normalized, rank-centric metric. We define the reward for the $i$-th response with rank $r_i$ as:

$$s_i^{\text{rank}} = f_{\text{PRR}}\big(s_i^{\text{rule}}, r_i\big) = \frac{r_{\max} - r_i}{r_{\max} - 1}. \quad (4)$$

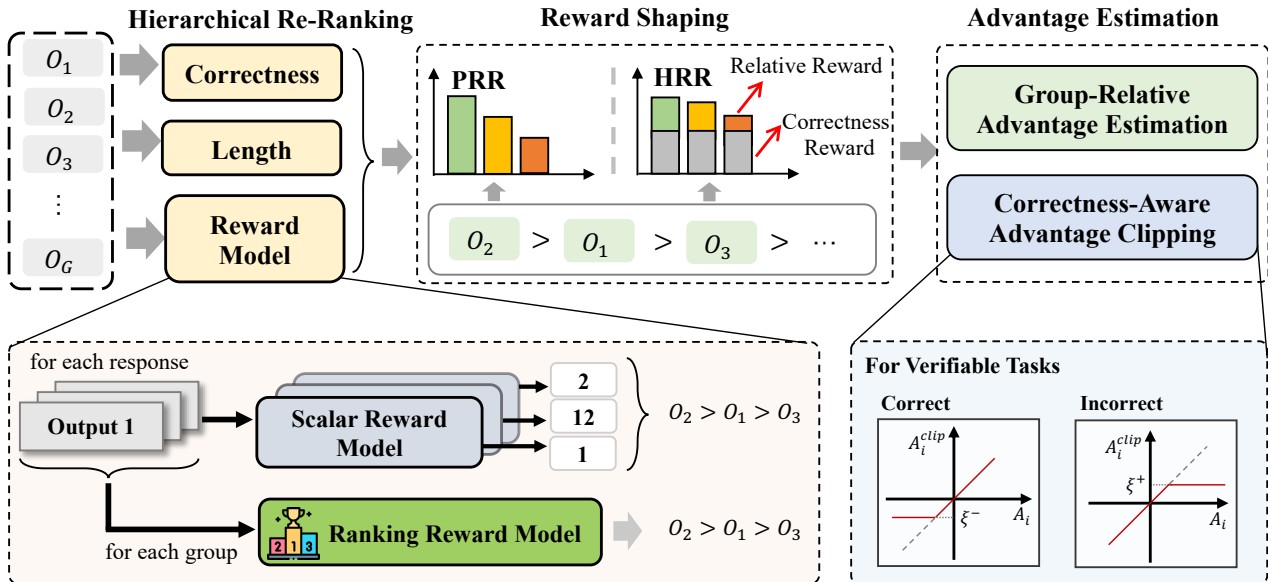

*Figure 2.* Overview of RLRR. The framework derives intra-group preference rankings based on Reward Model outputs, while incorporating correctness and length constraints. It then integrates relative rewards using either HRR or PRR, contingent on the availability of rule-based correctness rewards. Finally, the advantage estimation process accounts for correctness consistency and applies a clipping mechanism to handle contradictory samples.

This linear mapping projects the intra-group ranking into a fixed $[0, 1]$ interval. By decoupling the reward signal from the absolute magnitude of model outputs, PRR ensures that advantage estimation depends solely on relative ordering, thereby stabilizing training dynamics against the inherent score shifting of reward models.

**Correctness-Aware Advantage Clipping.** While relative advantages enable fine-grained learning, they may occasionally diverge from absolute correctness. A critical misalignment occurs when a valid solution receives a negative advantage simply because it ranks lower within a high-performing group. To prevent the policy from being penalized for generating correct outputs, we introduce a clipping strategy for the raw advantage $A_i$:

$$A_i^{\text{clip}} = \begin{cases} \max(A_i, \xi^-), & \text{if } o_i \text{ is correct,} \\ \min(A_i, \xi^+), & \text{if } o_i \text{ is incorrect,} \end{cases} \quad (5)$$

where $\xi^-$ and $\xi^+$ serve as safety margins. This mechanism restricts the magnitude of penalties for valid but suboptimal responses, ensuring that the model learns to differentiate quality nuances without compromising its fundamental capability to generate correct outcomes due to excessive discouragement.

### 4.2. Ranking Reward Model

SRMs typically employ a pointwise scoring approach, evaluating responses in isolation to assign absolute scalar values. Lacking comparative context, these independent scores may

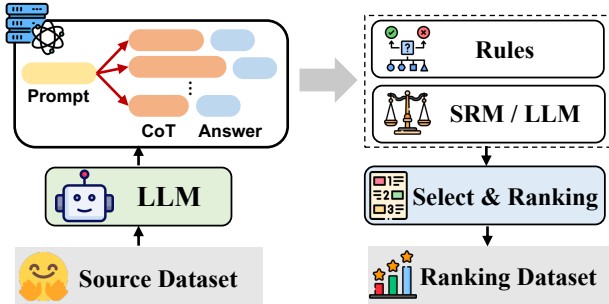

*Figure 3.* Training-data processing pipeline for the Ranking Reward Model, where responses are scored by rule-based rewards and an SRM, then ranked and shuffled to mitigate positional bias.

result in inaccurate rankings when converted to an order. To address this, we introduce the **Ranking Reward Model (Ranking RM)**. Unlike SRMs, the Ranking RM accepts a list of responses as a collective input and directly outputs their relative ordering. By processing candidates within a shared context, it yields a robust ranking signal that is more reliable than sorting independent scalar scores.

We instantiate the Ranking RM using a pretrained LLM backbone equipped with a classification head to predict ranking permutations, optimized via cross-entropy loss. To construct high-quality training data, we follow the pipeline illustrated in Figure 3. Specifically, we establish a hierarchical ranking structure where responses with verified correct outcomes strictly outrank incorrect ones. To determine the relative order of responses with identical correctness labels,

we primarily utilize scores from an SRM. However, if the SRM contradicts the ground truth by assigning higher scores to incorrect responses than to correct ones, we discard the unreliable SRM signal and employ a stronger LLM to derive the final rankings. This fallback is invoked only in rare cases; the vast majority of training instances rely solely on rule-based verification and SRM scores.

### 4.3. Hierarchical Re-ranking

Initial rankings derived from reward models may occasionally diverge from ground truth verification. To rectify these potential misalignments, we implement a lexicographical re-ranking strategy to strictly align model predictions with correctness and conciseness constraints. This mechanism primarily prioritizes rule-based correctness to ensure correct solutions outrank incorrect ones, followed by length re-ranking to counteract the verbosity bias inherent in reward models and favor concise responses. We formalize this ranking objective using a coarse-grained discretization function:

$$\mathcal{B}_i = \begin{cases} \left\lfloor \dfrac{\ell_i}{\lambda} \right\rfloor, & \text{if } o_i \text{ is correct,} \\ +\infty, & \text{if } o_i \text{ is incorrect,} \end{cases} \quad (6)$$

where $\ell_i$ represents the length of response $o_i$ and $\lambda$ serves as a hyperparameter controlling the bin granularity. Responses are sorted in ascending order of $\mathcal{B}_i$. Consequently, among correct responses falling within the same length bin, the final relative order follows the original preference predicted by the reward model. This hierarchical strategy guarantees that the final reward signal aligns with validity and efficiency constraints while preserving the fine-grained quality distinctions captured by the reward model.

---

**Algorithm 1** RLRR on Verifiable Task
___
**Input** policy $\pi_\theta$, dataset $\mathcal{D}$, rule-based verifier $R_\phi$, reward model $R_\psi$, group size $G$.
  **for** step = 1, ..., M **do**
    Sample a batch $\mathcal{D}_b$ from $\mathcal{D}$ and set $\pi_{\theta_{\text{old}}} \leftarrow \pi_\theta$
    **for** each $q \in \mathcal{D}_b$ **do**
      **Rollout:** Sample a group of responses $\{o_i\}_{i=1}^G \sim \pi_{\theta_{\text{old}}}(q)$.
      **Reward Calculation:** Compute correctness scores $s^{\text{rule}}$ and raw relative ranks $r^{\text{raw}}$ using $R_\phi$ and $R_\psi$.
      **Hierarchical Re-ranking:** Derive global ranks $r_i$ by lexicographically sorting the tuple $(s^{\text{rule}}, \mathcal{B}, r^{\text{raw}})$.
      **Reward Shaping:** Compute shaping rewards $s^{\text{rank}}$ using the rank-mapping function $f(s^{\text{rule}}, r)$.
      **Advantage Estimation:** Compute group advantages and apply correctness-aware clipping via Equation (5).
      **Policy update:** Update the policy $\pi_\theta$ by maximizing the GRPO objective (Equation (2)).
    **end for**
  **end for**
**Output:** The final policy $\pi_\theta$

---

Since the Ranking RM evaluates $n$ responses simultaneously, we set the GRPO group size $G$ as a multiple of $n$ and partition the responses into $G/n$ subgroups. As illustrated in Figure 2, we apply the Ranking RM to each subgroup to generate local raw ranks $r_i^{\text{raw}}$ and then perform the hierarchical re-ranking to derive the final global ranks $r_i$ for all $G$ responses. The complete RLRR is formalized in Algorithm 1. This integration of intra-group relative ordering effectively mitigates gradient vanishing induced by sparse rewards and circumvents the reliability limitations of absolute scoring. Furthermore, it facilitates priority-aware multi-objective optimization by synthesizing correctness, efficiency, and reasoning quality into a unified ranking signal.

## 5. Experiments

To evaluate the effectiveness of RLRR and the Ranking RM, we conduct experiments across three distinct dimensions:

- **Verifiable Reasoning Tasks**: We assess performance on mathematical and logical reasoning benchmarks where explicit ground truth is available for verification.

- **Open-ended Writing Tasks**: We utilize Writing-Bench (Wu et al., 2025) to evaluate generation quality across a diverse range of domains and writing styles.

- **Reward Model Evaluation**: We benchmark the discriminative capability of the Ranking RM using reward-guided test-time scaling (Zou et al., 2025).

### 5.1. Verifiable Reasoning Tasks

**Baselines.** We conduct our experiments on DeepSeek-R1-Distill-Qwen-1.5B and DeepSeek-R1-Distill-LlaMA-8B (Guo et al., 2025). Our primary comparison is against four recent state-of-the-art reinforcement learning methods: (1) GRPO (Shao et al., 2024), (2) Dr.GRPO (Liu et al., 2025b), (3) GPG (Chu et al., 2025), and (4) DAPO (Yu et al., 2025). The Ranking RM is trained on 25k data using Qwen2.5-7B-Instruct-1M (Team, 2025). We denote our method's two variants as RLRR(H) and RLRR(P), corresponding to the Hybrid Relative Reward and Pure Relative Reward strategies, respectively.

**Datasets.** For RL training, we use approximately 16,000 mathematics and logic samples filtered by difficulty from the GURU dataset (Cheng et al., 2025), and for the 8B model, we additionally include the Open-RS dataset (Dang & Ngo, 2025). For ablation and analysis experiments, we train the 1.5B model using the SimpleRL dataset (Zeng et al., 2025). For evaluation, we employ several challenging mathematical and logical reasoning benchmarks to assess our models' performance. Detailed descriptions and references for all evaluation datasets are provided in Appendix A.2.

**Performance.** As presented in Table 1, RLRR demonstrates

*Table 1.* Overall performance on eight competition-level mathematical reasoning benchmarks and two logic reasoning benchmarks. **Bold** and underlined indicate the best and second-best performance, respectively.

| | METHOD | MATHEMATICAL REASONING | | | | | | | | | LOGIC REASONING | | | Avg Len. |
|---|---|---|---|---|---|---|---|---|---|---|---|---|---|---|
| | | AIME 24 | AIME 25 | MATH 500 | GSM8K | Olympiad | GaoKao | Minerva | AMC | Avg | Zebra | Ordering | Avg | |
| | | *Avg@32* | *Avg@32* | *Avg@4* | *Avg@4* | *Avg@4* | *Avg@4* | *Avg@4* | *Avg@16* | | *Avg@4* | *Avg@4* | | *Tokens* |
| *DeepSeek-Qwen-1.5B* | Baseline | 26.4 | 21.8 | 83.2 | 86.1 | 41.7 | 71.3 | 26.3 | 61.4 | 52.3 | 0.7 | 14.0 | 7.4 | 10050 |
| | GRPO | 29.3 | 23.4 | 82.7 | 85.7 | 42.8 | 72.7 | 27.6 | 63.0 | 53.4 | 2.5 | 22.4 | 12.5 | 6943 |
| | Dr.GRPO | 29.1 | 23.8 | 83.0 | 85.6 | 43.7 | 73.6 | 27.0 | 63.6 | 53.7 | 3.0 | 20.4 | 11.7 | 7025 |
| | DAPO | 28.1 | 22.9 | 83.5 | 86.3 | 44.1 | 73.7 | 28.0 | 65.3 | 54.0 | 6.8 | 28.4 | 17.6 | 9536 |
| | GPG | 30.4 | 24.2 | 84.2 | 86.4 | 44.2 | 73.4 | **28.3** | 62.8 | 54.2 | 3.8 | 22.2 | 13.0 | 8548 |
| | RLRR(H) | 31.1 | **24.5** | **84.6** | **86.5** | 47.0 | **74.6** | 28.0 | 66.8 | 55.4 | **10.1** | **39.5** | **24.8** | **6484** |
| | RLRR(P) | **32.5** | 23.2 | 83.8 | 86.2 | **47.8** | 74.4 | 28.1 | **68.3** | **55.5** | 6.1 | 31.4 | 18.8 | 6727 |
| *DeepSeek-LLaMA-8B* | Baseline | 46.8 | 29.3 | 88.4 | 91.0 | 51.6 | 79.3 | 28.9 | 77.3 | 61.6 | 9.0 | 55.8 | 32.4 | 7871 |
| | GRPO | 48.1 | 30.0 | 88.9 | 91.3 | 53.1 | 80.4 | 31.3 | 80.2 | 62.9 | 30.8 | 77.7 | 54.3 | 6567 |
| | Dr.GRPO | 48.8 | 29.8 | 89.4 | 91.5 | 53.5 | 79.0 | 30.8 | 80.9 | 63.0 | 23.5 | 74.5 | 49.0 | 6466 |
| | DAPO | 46.8 | 34.2 | 89.3 | **92.1** | 58.4 | 81.9 | 32.0 | 79.0 | 64.2 | 37.3 | **89.3** | 63.3 | 5871 |
| | GPG | 47.6 | 30.1 | 89.0 | 91.3 | 54.1 | 80.4 | 31.3 | 80.2 | 63.0 | 24.6 | 78.8 | 51.7 | 6920 |
| | RLRR(H) | **50.5** | **35.4** | **91.4** | **92.1** | **59.0** | **82.1** | **33.5** | **83.0** | **65.9** | **40.4** | 89.2 | **64.8** | **5025** |
| | RLRR(P) | 48.8 | 34.1 | 89.4 | 91.7 | 54.8 | 80.5 | 32.4 | 80.9 | 64.1 | 35.8 | 88.0 | 61.9 | 5287 |

robust performance on both mathematical and logical reasoning benchmarks, with both the HRR and PRR variants yielding effective results. Notably, beyond achieving high accuracy, RLRR maintains superior inference efficiency, requiring the lowest token consumption among all baseline methods. This efficiency underscores the efficacy of ranking-based optimization in balancing performance with computational cost. Specifically, HRR, which anchors advantage estimation to ground truth rules, yields the most substantial and stable improvements on verifiable tasks. In contrast, PRR achieves slightly lower performance, a result attributed to its exclusion of direct rule-based supervision. Collectively, these findings validate the effectiveness of RLRR in enhancing both mathematical and logical reasoning capabilities across diverse domains.

### 5.2. Open-ended Writing Tasks

**Datasets and Models.** Following the pipeline in Figure 3, we constructed the preference dataset by generating ranked responses for 10k Dolphin-R1[1] samples using models of varying scales. We then fine-tuned the Qwen3-1.7B backbone (Yang et al., 2025a) on a separate set of 22k samples.

**Baselines.** We adopt GRPO (Shao et al., 2024) as the algorithmic baseline for comparison. In addition, we benchmark the Ranking RM against two state-of-the-art reward models: (1) Skywork-Reward-V2-Llama-3.1-8B (Liu et al., 2025a),

(2) URM-LLaMa-3.1-8B (Lou et al., 2024). We also apply RLRR with PRR on top of these SRMs to convert their absolute scores into relative rankings.

**Performance.** Table 2 presents the performance of RLRR on open-ended tasks. By converting absolute scalar scores from two distinct reward models into relative preferences through the PRR approach, we achieve improvements across most domains. Open-ended tasks lack rule-based verifiers, making the SRM the sole source of reward signal. Under such conditions, the score instability of SRMs directly propagates into GRPO's advantage estimation, whereas PRR neutralizes this instability by projecting scores onto a fixed ordinal scale. This highlights the effectiveness of relative ranking in enhancing both the stability and performance of GRPO. Moreover, when RLRR is paired with the fine-tuned Ranking RM, it achieves the best results in the majority of domains, underscoring that evaluating the preference order of multiple responses provides a more robust learning signal than scoring individual responses. These findings reinforce the value of ranking-based approaches for improving performance in open-ended tasks.

### 5.3. Evaluation of Ranking Reward Model

Figure 4 compares the SRM Skywork-Reward-V2-Llama-3.1-8B (Liu et al., 2025a) and the Ranking RM under two experimental configurations. Figure 4a demonstrates their performance on mathematical reasoning tasks using the DeepSeek-LlaMA-8B (Guo et al., 2025), where we sample

---

[1] https://huggingface.co/datasets/QuixiAI/dolphin-r1

*Table 2.* Overall performance on Writing Bench, comparing reward scores and reward ranks. The '↪' symbol indicates that RLRR with PRR is applied using the reward model listed above. **Bold** and underlined mark the best and second-best results, respectively.

| METHOD | OVERALL | ACADEMIC & ENGINEERING | FINANCE & BUSINESS | POLITICS & LAW | LITERATURE & ARTS | EDUCATION | ADVERTISING & MARKETING |
|---|---|---|---|---|---|---|---|
| Qwen3-1.7B | 70.06 | 72.60 | 71.17 | 70.99 | 63.22 | 73.52 | 70.27 |
| SFT | 70.90 | 73.17 | 70.89 | 71.47 | 65.75 | 74.68 | 71.09 |
| Skywork-8B | 72.88 | 74.56 | 72.81 | 72.40 | 69.68 | 76.00 | 73.42 |
| ↪ *RLRR* | 73.64 (▲0.76) | 75.25 (▲0.69) | 73.71 (▲0.90) | 73.77 (▲1.37) | 69.81 (▲0.13) | 77.22 (▲1.22) | 73.57 (▲0.15) |
| URM-8B | 73.12 | 75.14 | 73.65 | 73.47 | 69.06 | 76.16 | 72.25 |
| ↪*RLRR* | 74.71 (▲1.59) | 76.46 (▲1.32) | 75.47 (▲1.82) | 75.30 (▲1.83) | 70.86 (▲1.80) | 77.08 (▲0.92) | 73.72 (▲1.47) |
| Ranking RM(ours) | **81.33** | **83.27** | **82.92** | **81.68** | **75.90** | **84.16** | **80.96** |

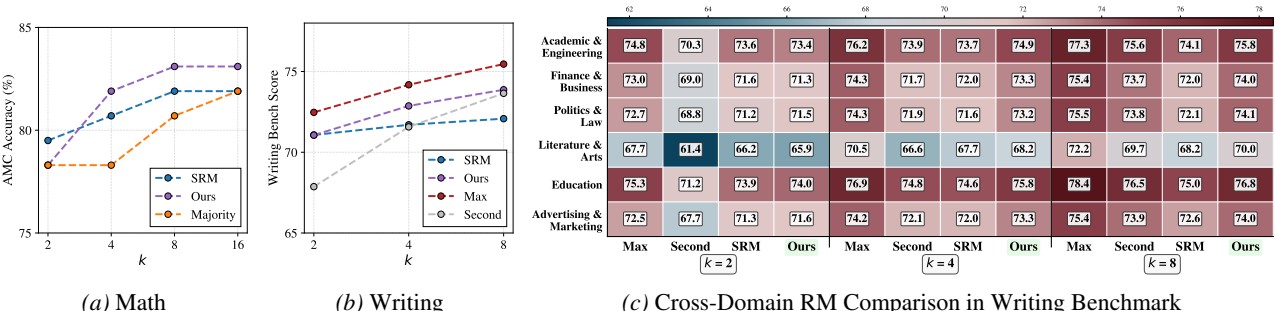

*(a)* Math    *(b)* Writing    *(c)* Cross-Domain RM Comparison in Writing Benchmark

*Figure 4.* Evaluation of reward models under Best-of-N test-time scaling. (a) Best-of-N scaling curves on the Math benchmark. (b) Best-of-N scaling curves on the Writing benchmark. (c) Cross-domain reward model comparison on the Writing benchmark.

$k$ responses per prompt and select via either SRM or Ranking RM with Majority voting. Results show Ranking RM achieves comparable or superior accuracy to SRM, with the performance gap widening as $k$ increases. Figure 4b and Figure 4c illustrate Ranking RM's superior performance in writing tasks evaluated on Qwen3-1.7B (Yang et al., 2025a), where we sample eight responses per instance. The *Second* designation denotes the second-highest-scoring sample. Ranking RM consistently selects higher-quality responses, explaining its significant improvement in compositional tasks. We attribute these improvements to Ranking RM's focus on relative intra-group quality assessment rather than absolute scoring. Notably, Ranking RM demonstrates strong generalization across domains despite limited training data.

## 5.4. Method Analysis

We analyze RLRR from multiple perspectives, with detailed results provided in Appendix C.

**Impact of Dataset Difficulty.** The dataset difficulty directly affects the proportion of the effective prompts. To examine this, we fine-tune a 1.5B model on three levels: (1) Easy: GSM8k (Cobbe et al., 2021), (2) Medium: SimpleRL (Zeng et al., 2025), and (3) Hard: Open-RS (Dang & Ngo, 2025). Figure 5 presents the performance of GRPO and RLRR across datasets of varying difficulty. We observe that the

fraction of effective prompts under GRPO remains relatively low and exhibits difficulty dependent dynamics, while RLRR maintains full utilization of group-wise information regardless of task complexity. The underlying mechanism varies across difficulty levels. On easy data, most groups become unanimously correct early in training, rapidly depleting GRPO's gradient signal. RLRR continues to learn by extracting fine-grained quality distinctions among correct responses via the Ranking RM. On hard data, groups are initially unanimously incorrect and gradually yield mixed outcomes as the policy improves. RLRR further accelerates this process by ranking responses that are closer to correctness even within all-incorrect groups. Across all difficulty levels, RLRR consistently outperforms GRPO, suggesting that relative ranking relaxes the dependency on carefully calibrated data difficulty by extracting useful signals even from prompts that would be uninformative under absolute scoring. Further analysis about the impact of dataset difficulty is provided in Appendix C.2.

**Ablation of Hierarchical Re-ranking.** We analyze the contribution of different components in the re-ranking stage. Table 3 shows that removing correctness consistently degrades performance, confirming its necessity for reliable reasoning. Interestingly, relaxing the length constraint slightly improves results on several datasets, as aggressive length penalization may restrict the model's exploration of diverse

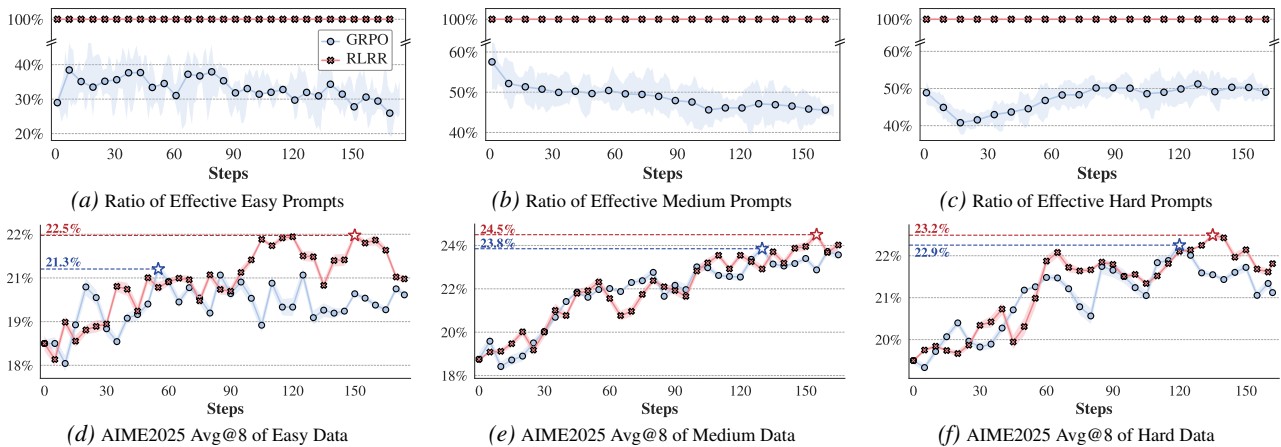

*Figure 5.* Comparison of GRPO and RLRR across different data difficulty levels. Top: ratio of effective prompts for easy, medium, and hard data. Bottom: AIME2025 Avg@8 performance on the corresponding difficulty subsets.

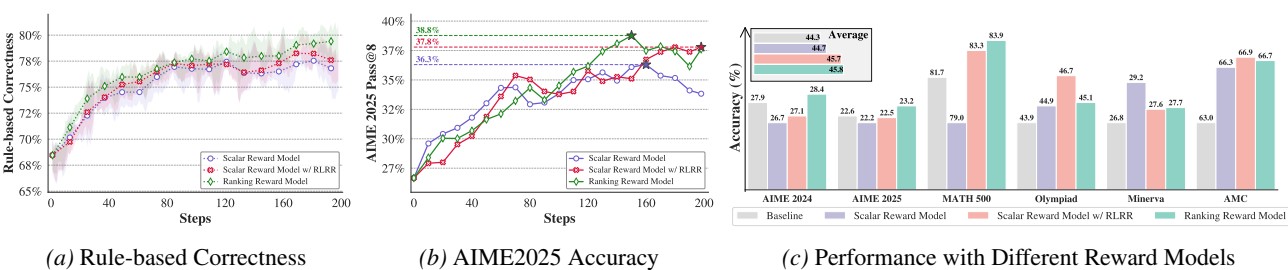

*(a)* Rule-based Correctness     *(b)* AIME2025 Accuracy     *(c)* Performance with Different Reward Models

*Figure 6.* Training dynamics and reward analyses of RLRR excluding correctness rewards. (a) Rule-based correctness reward trajectory during training without explicit correctness supervision. (b) AIME2025 accuracy evolution over training steps. (c) Performance comparison across different reward model configurations on mathematical reasoning benchmarks.

*Table 3.* Ablation study results on Hierarchical Re-ranking. **Bold** marks the best result in each column. *Cor.* means Correctness, and *Len.* means Length.

| METHOD | AIME24 | AIME25 | MATH | Olym. | Mine. | AMC | Avg | Len. |
|---|---|---|---|---|---|---|---|---|
| RLRR | 30.8 | 24.5 | 83.8 | 46.5 | 27.6 | **69.1** | 47.1 | 6013 |
| w/o *Cor.* | 28.4 | 23.1 | 83.9 | 44.8 | 27.7 | 66.3 | 45.7 | **5720** |
| w/o *Len.* | **31.3** | **24.7** | **85.0** | **47.2** | **27.9** | 66.8 | **47.2** | 6317 |

*Table 4.* Performance comparison of RLRR across different group-based reinforcement learning algorithms. *w/ RLRR* denotes applying RLRR on top of the base algorithm in the preceding row.

| METHOD | AIME24 | AIME25 | Minerva | Olympiad | AMC | Avg |
|---|---|---|---|---|---|---|
| CISPO | 27.7 | 22.8 | 26.6 | 42.3 | 61.3 | 36.1 |
| *w/ RLRR* | 29.8 (▲2.1) | 23.4 (▲0.6) | 27.5 (▲0.9) | 42.2 (▼0.1) | 62.7 (▲1.4) | 37.1 (▲1.0) |
| RLOO | 28.1 | 22.4 | 26.3 | 42.1 | 61.4 | 36.1 |
| *w/ RLRR* | 28.9 (▲0.8) | 23.2 (▲0.8) | 27.4 (▲1.1) | 43.2 (▲1.1) | 62.9 (▲1.5) | 37.1 (▲1.0) |

reasoning strategies, particularly on problems that require multi-step deduction. Nevertheless, we retain the length re-ranking mechanism because the observed gains from relaxing it remain marginal, while the constraint provides tangible benefits in robustness under response length restrictions and reduced inference cost. Further discussion on the length re-ranking mechanism is provided in Appendix C.4. Overall, correctness serves as the primary factor for stabil-

ity, while length control requires a careful balance between conciseness and expressiveness.

**Comparison of Reward Shaping Strategies.** We further investigate strategies for using reward models. As shown in Figure 6c, SRMs suffer from instability, occasionally leading to performance degradation on specific benchmarks. In contrast, converting absolute scores into relative rankings significantly stabilizes the reward signal and yields consistent performance gains. Figure 6a and Figure 6b illustrate this effect from the perspective of training dynamics, highlighting how relative rewards mitigate signal fluctuations. Furthermore, adopting our proposed Ranking RM delivers the most substantial improvements, providing empirical evidence that listwise ranking signals offer superior reliability compared to pointwise scalar values. These findings confirm that relative ranking-based rewards effectively enhance both training stability and final model performance.

**Generalizability to Other Group-based Algorithms.** To validate the versatility of our framework, we investigate the impact of relative rewards on two additional group-based reinforcement learning algorithms: CISPO (Chen et al., 2025) and RLOO (Ahmadian et al., 2024). As detailed in Table 4, integrating relative rewards yields consistent

*Table 5.* Overall performance on seven competition-level mathematical reasoning benchmarks of pre-trained model. **Bold** and underlined indicate the best and second-best performance, respectively.

| METHOD | REWARD | AIME24 | AIME25 | MATH500 | Olympiad | GaoKao | Minerva | AMC | Avg | Avg Len. |
|--------|--------|--------|--------|---------|----------|--------|---------|-----|-----|----------|
| Qwen-2.5-7B | - | 3.6 | 1.1 | 40.2 | 16.2 | 33.2 | 14.9 | 22.4 | 22.8 | 915 |
| GRPO | Absolute | 8.2 | 3.8 | 63.2 | 27.3 | 55.3 | 24.9 | 36.1 | 31.3 | 857 |
| DAPO | Absolute | 7.6 | 3.6 | 63.4 | 27.3 | 55.5 | 24.2 | 37.0 | 31.2 | 874 |
| RLRR w/ SRM | Relative | 8.6 | **3.9** | **63.9** | 27.4 | **56.0** | **25.8** | 37.5 | 31.9 | **827** |
| RLRR | Relative | **10.0** | 3.6 | 63.1 | **27.9** | 55.8 | 25.5 | **38.8** | **32.1** | 852 |

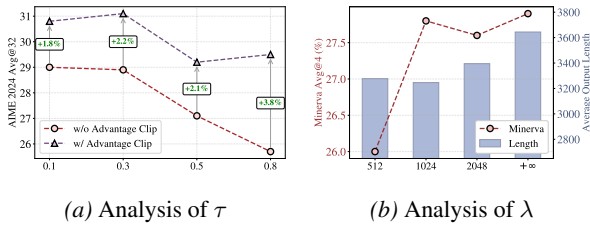

*(a) Analysis of $\tau$*      *(b) Analysis of $\lambda$*

*Figure 7.* Sensitivity Analysis of $\tau$, Advantage Clipping, and $\lambda$. $\lambda = +\infty$ means RLRR w/o Length Re-ranking.

performance improvements for both baselines. Although these algorithms differ in their specific implementations, they all estimate advantages based on the differences among multiple responses within a group, which is the key property that makes relative rewards effective, demonstrating the generalizability of RLRR framework.

**Evaluation on Pretrained Models.** We evaluate the performance of diverse methods using Qwen2.5-7B (Team, 2024) as the backbone, with detailed results summarized in Table 5. RLRR achieves the best performance across the majority of datasets, demonstrating particularly significant improvements on challenging benchmarks like AIME2024 and AMC. Regarding generation length, we observe minimal variation among different methods, which we attribute to the limited reasoning capacity inherent to the base model. Notably, RLRR yields strong results even when instantiated with an SRM. This finding suggests that the relative reward shaping strategy serves as a critical driver of the observed performance gains, validating its effectiveness in extracting robust training signals even from scalar models.

**Hyperparameter Sensitivity.** We examine the impact of $\tau$ and $\lambda$. As shown in Figure 7a, excessive $\tau$ dilutes rule-based rewards, potentially misaligning gradients with correctness and degrading performance. Introducing advantage clipping effectively mitigates this instability. As illustrated in Figure 7b, relaxing the conciseness constraint through $\lambda$ leads to longer responses. Notably, the model retains high accuracy even at reduced lengths, confirming that the length re-ranking effectively trims redundancy. This finding aligns with the results in Table 3, further corroborating the effectiveness of length constraints in reducing redundancy.

### 5.5. Key Findings of Experiments

We synthesize the principal insights emerging from our experimental evaluation. Across diverse tasks, backbone models, and algorithmic variants, a consistent picture emerges: replacing absolute reward scores with relative rankings yields robust and transferable benefits. The findings below highlight the most salient conclusions.

> **Key Findings**
> 1. Relative rankings consistently improve performance on both verifiable reasoning and open-ended generation.
> 2. The gains do not depend on the Ranking RM; converting SRM scores into relative rankings already yields substantial improvements.
> 3. RLRR generalizes beyond GRPO to other group-based algorithms including CISPO and RLOO.
> 4. RLRR maintains effective learning signals on easy data where GRPO suffers from gradient depletion.

## 6. Conclusion

In this paper, we introduce RLRR, a framework that shifts the paradigm of group-based optimization from absolute scoring to relative preference ranking. By synthesizing intra-group comparisons through relative reward shaping, RLRR effectively mitigates gradient vanishing caused by sparse supervision and resolves the optimization instability inherent in SRMs. Extensive experiments demonstrate that RLRR yields consistent improvements across verifiable reasoning and open-ended writing tasks. These findings underscore the superiority of relative preference signals over absolute scoring for model optimization. Furthermore, we show that the Ranking RM serves as a robust listwise evaluator that naturally aligns with the comparative structure of group-based learning, delivering effective signals even with limited training data. One limitation of our approach is that the Ranking RM increases input context length proportionally to group size, which may constrain scalability to very large groups or very long responses. Future work will explore more efficient listwise ranking mechanisms to alleviate this constraint.

## Acknowledgements

This work is supported by the National Natural Science Foundation of China under Grant 62376194, and in part by China Scholarship Council Grant 202406250137.

## Impact Statement

This paper presents work whose goal is to advance the field of Machine Learning, particularly in the domain of reinforcement learning for Large Language Models. Our research aims to improve the reliability of models in mathematical reasoning and open-ended generation tasks. There are many potential societal consequences of advancing LLM capabilities, none of which we feel must be specifically highlighted here beyond the general discourse on AI safety and alignment.

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

## A. Training Details

### A.1. Settings

We cap the generated output length at 8,192 tokens and form groups of size $G = 8$ per prompt. For the Ranking RM, we set the sortable subset size to $n = 4$. Unless otherwise noted, hyperparameters are fixed as follows: $\lambda = 2048$, $\xi^+/\xi^- = \pm 10^{-3}$, $\tau = 0.1$, and sampling temperature $T = 1.0$ during data collection. Our method and all baselines are implemented on top of the VeRL (Sheng et al., 2025) framework.

For the reward model evaluation, we set $n = 4$. When $n = 2$, we repeat the process once for each sample. For $n = 8$ or $n = 16$, we first divide the samples into multiple groups of size 4, then select the best from each group. Afterward, we continue with the Ranking RM for a second round of selection until the optimal answer is chosen.

### A.2. Evaluation Datasets

We evaluate our models on seven mathematical reasoning benchmarks: Math500 (Hendrycks et al., 2021; Lightman et al., 2023), AIME24 (Art of Problem Solving, 2024a), AIME25 (Art of Problem Solving, 2025), AMC (Art of Problem Solving, 2024b), Minerva Math (Lewkowycz et al., 2022), Gaokao (Zhang et al., 2023), and Olympiad Bench (He et al., 2024), which cover a broad range of mathematical difficulty and problem types. For logical reasoning, we select two representative benchmarks: Zebra Puzzle (Cheng et al., 2025), and Ordering Puzzle (Cheng et al., 2025). These datasets are widely recognized and present diverse challenges for evaluating both mathematical and general reasoning abilities. We configure the maximum generation length to 16,384 tokens for mathematical reasoning tasks and 32,768 tokens for logical reasoning tasks. Conversely, for open-ended writing tasks, we restrict the generation limit to 8,192 tokens and utilize the open-source Critic Model (Wu et al., 2025) as the judge.

## B. Theoretical Analysis of Relative Reward Stability

In group-based reinforcement learning frameworks such as GRPO, policy optimization relies fundamentally on the accurate estimation of the advantage function. This estimation requires a robust baseline derived from the group statistics. However, Scalar Reward Models (SRM) introduce instability due to the unbounded nature of their output scores. In this section, we provide a theoretical analysis demonstrating that the loss function of SRMs inherently drives rewards toward divergence, propagating high variance into the advantage estimation. We then formally prove that the proposed RLRR method mitigates this issue by imposing strict variance bounds through reward normalization.

### B.1. Gradient-Driven Divergence in Scalar Reward Models

To understand the source of score instability, we analyze the optimization dynamics of the standard reward modeling objective. SRMs typically parameterize the preference probability using the Bradley-Terry model. Given a prompt $x$, a preferred response $y_w$, and a rejected response $y_l$, the preference probability is defined as $\sigma(\delta)$, where $\sigma$ is the sigmoid function and $\delta = r(x, y_w) - r(x, y_l)$ is the reward difference.

The training objective minimizes the negative log-likelihood over the dataset $\mathcal{D}$:

$$\mathcal{L}_{\text{SRM}} = -\mathbb{E}_{\mathcal{D}}\left[\log \sigma(\delta)\right] = \mathbb{E}_{\mathcal{D}}\left[\log(1 + e^{-\delta})\right] \tag{7}$$

By examining the gradient of this loss function with respect to the reward difference $\delta$, we reveal the driving force behind the unbounded scores. The derivative is given by:

$$\frac{\partial \mathcal{L}}{\partial \delta} = \frac{-e^{-\delta}}{1 + e^{-\delta}} = \sigma(\delta) - 1 \tag{8}$$

Since the image of the sigmoid function for any finite input is the open interval $(0, 1)$, the gradient $\sigma(\delta) - 1$ is strictly negative for all finite $\delta$. This indicates that $\mathcal{L}_{\text{SRM}}$ is monotonically decreasing with respect to $\delta$. Consequently, the infimum of the loss is approached only as $\delta \to +\infty$.

During optimization, the gradient descent process exerts a constant pressure to maximize the gap between $y_w$ and $y_l$. In the absence of explicit regularization on the reward magnitudes, this creates a runaway effect where the absolute values

of $r(x, y)$ must grow indefinitely to satisfy the objective. Furthermore, the Bradley-Terry model is invariant to translation, meaning $r(x, y)$ and $r(x, y) + C$ yield identical losses. This lack of anchoring, combined with the gradient pressure for divergence, results in a reward distribution with theoretically unbounded support. This mathematical property directly leads to the generation of extreme values and high variance during the RL training phase.

### B.2. Bias Propagation in Group-Based Advantage Estimation

The unbounded nature of SRM outputs significantly impacts the stability of group-based estimators used in algorithms like GRPO. The advantage $A_i$ for the $i$-th sample in a group of size $G$ is computed by normalizing the reward score $s_i$ against the group statistics:

$$A_i = \frac{s_i - \bar{s}}{\sigma_G} \tag{9}$$

Here $\bar{s}$ denotes the sample mean and $\sigma_G$ denotes the sample standard deviation. The reliability of $A_i$ depends on $\bar{s}$ being a robust estimator of the true distributional mean. However, as established in the previous subsection, the SRM tends to produce heavy-tailed or extreme score distributions.

Consider a scenario where the SRM assigns an extreme outlier score to a single sample due to model overconfidence or the divergence mechanism described above. Since the sample mean $\bar{s}$ is not robust to outliers, this single value shifts the baseline significantly. Consequently, the advantages computed for the remaining "normal" samples are skewed, often collapsing towards zero or exhibiting incorrect signs. The high dispersion of the scalar scores implies that while the ranking might be preserved, the magnitude of the advantage becomes dominated by numerical noise rather than signal quality. This variance in the baseline estimator introduces bias into the policy gradient update, thereby destabilizing the training process.

### B.3. Variance Bounds via RLRR

The proposed Reinforcement Learning with Relative Reward (RLRR) framework resolves this instability by fundamentally altering the support of the reward distribution. Instead of relying on raw scalar outputs, RLRR enforces constraints that map rewards into a compact set.

As defined in the method section, the Pure Relative Reward maps rankings linearly to the interval $[0, 1]$, while the Hybrid Relative Reward utilizes the hyperbolic tangent function to constrain corrections within a fixed range derived from the hyperparameter $\tau$. In both cases, the reward $R$ is a random variable strictly bounded within a finite interval $[a, b]$.

We invoke Popoviciu's inequality on variances to demonstrate the theoretical benefit of this design. For any random variable $X$ with support bounded in $[a, b]$, the variance is strictly limited by:

$$\text{Var}(X) \le \frac{(b-a)^2}{4} \tag{10}$$

For PRR, where the range is $[0, 1]$, the variance of the reward distribution is strictly upper-bounded by $0.25$, regardless of the model confidence or training duration. Similarly, for HRR, the variance is bounded by a function of $\tau$.

By imposing this hard mathematical limit on variance, RLRR ensures that no single sample can exert an arbitrarily large influence on the group mean $\bar{s}$ or standard deviation $\sigma_G$. This guarantees that the advantage estimator remains robust even in the presence of distinct preference differences. Therefore, the transition from unbounded scalar rewards to bounded relative rewards provides a theoretical guarantee of stability for group-based policy optimization.

## C. Supplementary Results

### C.1. Absolute Baseline.

In our experiments, we compared the impact of using relative baselines (intra-group mean) versus absolute correctness baselines (for GRPO, a baseline of 0.5; for RLRR, a baseline of 1) on performance and stability. The results, shown in Figure 8a, indicate that using the absolute correctness baseline leads to a significant drop in performance. Figure 8b and 8d further reveal the instability introduced by the absolute baseline, particularly from the perspective of truncation

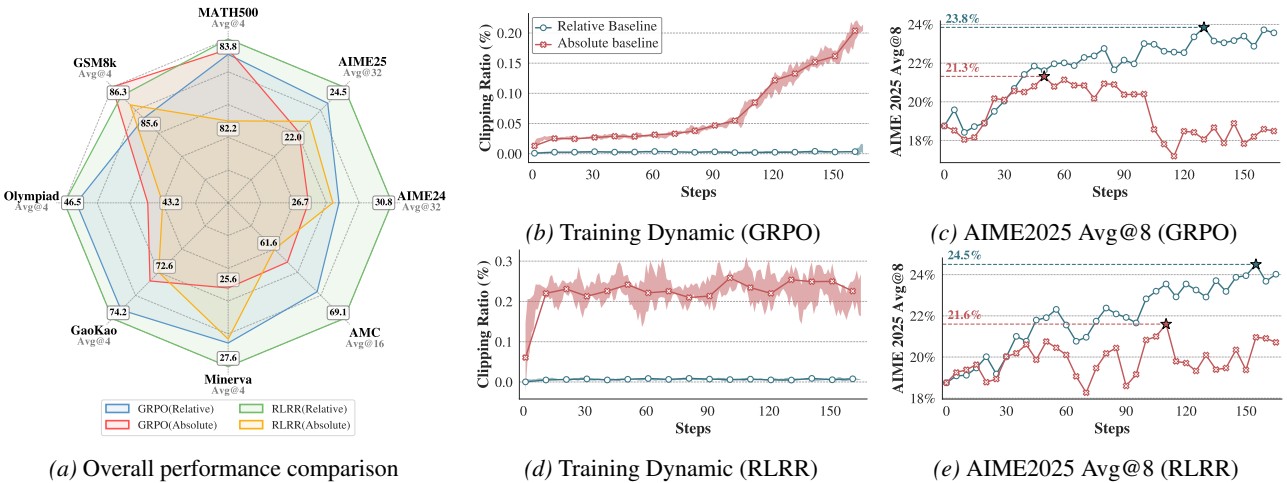

*(a)* Overall performance comparison     *(b)* Training Dynamic (GRPO)     *(c)* AIME2025 Avg@8 (GRPO)

*(d)* Training Dynamic (RLRR)     *(e)* AIME2025 Avg@8 (RLRR)

*Figure 8.* Effect of Absolute vs. Relative Baselines on GRPO and RLRR.

*Table 6.* Performance comparison across datasets of varying difficulty; **bold** indicates the best result.

| METHOD | | AIME24 | AIME25 | MATH500 | GSM8k | Olympiad | GaoKao | Minerva | AMC | Avg |
|---|---|---|---|---|---|---|---|---|---|---|
| Easy | GRPO | 28.5 | 22.8 | 82.6 | **87.0** | 44.0 | 72.3 | **28.2** | 62.7 | 53.5 |
| | DAPO | **30.0** | **22.9** | 82.4 | 86.0 | 42.6 | 73.0 | 26.3 | 62.3 | 53.2 |
| | RLRR | 29.7 | 22.8 | **83.2** | 86.8 | **44.1** | **74.6** | 27.4 | **64.5** | **54.1** |
| Medium | GRPO | 28.2 | 23.6 | 83.5 | 85.6 | 46.1 | 73.9 | 27.0 | 66.2 | 54.3 |
| | DAPO | 29.3 | 23.2 | 82.5 | 86.0 | 42.7 | 72.8 | 25.6 | 62.0 | 53.0 |
| | RLRR | **30.8** | **24.5** | **83.8** | **86.1** | **46.5** | **74.2** | **27.6** | **69.1** | **55.3** |
| Hard | GRPO | 27.9 | 23.6 | 82.9 | 85.9 | 43.4 | 74.2 | 26.3 | 65.4 | 53.7 |
| | DAPO | 28.1 | 22.1 | 82.9 | 85.8 | 43.3 | 72.6 | **27.4** | 62.2 | 53.0 |
| | RLRR | **28.8** | **23.8** | **83.6** | **86.2** | **45.0** | **74.9** | 26.7 | **66.4** | **54.4** |

rates. Additionally, Figure 8c and 8e demonstrate a decline in accuracy during the later stages of training, highlighting the unsuitability of the absolute baseline for long-term training.

### C.2. Impact of Dataset Difficulty

We analyze how dataset difficulty influences data efficiency and performance in Section 5.4. Table 6 reports a detailed comparison of three methods across difficulty levels. Moderate difficulty yields the best gains, whereas overly easy or overly hard data diminishes further improvement. In GRPO, extremes of difficulty tend to degenerate into *invalid prompts* that provide little learning signal and mainly act as a weak regularizer to prevent forgetting of trivial cases. In fact, on the easy subset many prompt groups are already close to unanimously correct at the beginning of training, so the GRPO effective prompt ratio starts at a relatively low level and quickly saturates. On the medium subset the GRPO effective prompt ratio decreases from about 60% at the beginning of training to about 40% near convergence, which is consistent with the global trend in Figure 1a. On the hard subset many prompt groups are initially unanimously incorrect and gradually become effective as the policy improves, which compensates for prompts that later turn unanimously correct and produces an almost flat curve. Across all difficulty levels, the absolute fraction of effective prompts under GRPO remains relatively low, indicating limited utilization of the available data. DAPO removes such invalid prompts altogether, which avoids noise but forfeits potential information contained therein. By contrast, RLRR leverages *all* samples by converting groupwise orderings into usable signal, thereby extracting additional knowledge even from otherwise low-value prompts. Figure 9 visualizes the fraction of effective data throughout RL training: RLRR maintains 100% effective utilization at all times, substantially exceeding the other methods and corroborating its advantages in both data efficiency and final performance.

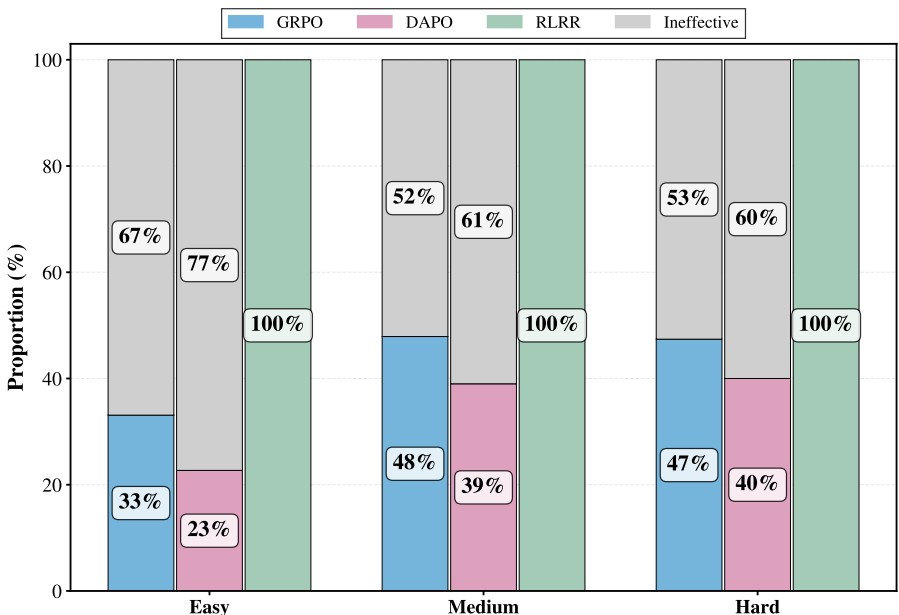

*Figure 9.* The proportion of effective data during the training phase for different methods.

*Table 7.* Performance Comparison Across Different Group Sizes

| $G$ | METHOD | MATHEMATICAL REASONING | | | | | | | | Avg |
|---|---|---|---|---|---|---|---|---|---|---|
| | | AIME24 | AIME25 | MATH500 | GSM8k | Olympiad | GaoKao | Minerva | AMC | |
| 4 | GRPO | 28.6 | 22.4 | 83.1 | 86.6 | 45.9 | 72.8 | 27.4 | 66.3 | 54.1 |
| | RLRR | 31.5 | 24.1 | 84.3 | 85.1 | 45.0 | 73.8 | 26.6 | 65.4 | 54.5 |
| 8 | GRPO | 28.2 | 23.6 | 83.5 | 85.6 | 46.1 | 73.9 | 27.0 | 66.2 | 54.3 |
| | RLRR | 30.8 | 24.5 | 83.8 | 86.1 | 46.5 | 74.2 | 27.6 | 69.1 | 55.3 |
| 16 | GRPO | 28.9 | 22.8 | 85.5 | 87.2 | 47.1 | 75.9 | 27.7 | 68.0 | 55.4 |
| | RLRR | 30.1 | 23.2 | 85.6 | 86.8 | 46.7 | 74.9 | 28.2 | 67.8 | 55.4 |

### C.3. Effect of Group Size on Method Performance

We conducted a comprehensive analysis of both methods' performance across varying group sizes $G$, as detailed in Table 7. The experimental results reveal that GRPO exhibits performance improvement with increasing $G$, primarily due to reduced occurrence of invalid groups at larger group sizes. In contrast, RLRR demonstrates consistent effectiveness even at smaller $G$ values through full utilization of all available training data. The experimental findings demonstrate that GRPO attains performance levels comparable to RLRR when operating with sufficiently large group sizes, as the increased sampling capacity enables more comprehensive data utilization.

### C.4. Effect of $\lambda$ on Method Performance

We analyze the sensitivity of model performance to the parameter $\lambda$ defined in Equation (6), as presented in Table 8. Relaxing the length constraint by increasing $\lambda$ leads to steady performance gains. When $\lambda$ is set to a large value, the constraint becomes negligible, yielding results comparable to the configuration without length re-ranking. On the other hand, given that the length constraint specifically targets correct responses, the model demonstrates resilience and maintains performance even under stricter constraints characterized by small $\lambda$ values.

In addition, we evaluate the robustness of the models by imposing varying maximum response length caps. While all

methods exhibit some performance degradation, the approach without length re-ranking suffers the most significant drop. This empirical evidence suggests that the baseline relies heavily on inherent verbosity to achieve correctness, whereas our length re-ranking mechanism successfully biases the model towards conciseness and ensures robustness against length restrictions.

*Table 8.* Sensitivity analysis of $\lambda$. Note that $\lambda = +\infty$ corresponds to RLRR without length re-ranking. MRL denotes the Max Response Length imposed during testing. Values in parentheses indicate the performance change relative to the 32k baseline. Green signifies no performance loss (optimal), gray indicates a moderate decline, and red highlights the most significant degradation.

| $\lambda$ | MRL | MATHEMATICAL REASONING | | | | | | | | | Avg Len |
| --- | --- | --- | --- | --- | --- | --- | --- | --- | --- | --- | --- |
| | | AIME24 | AIME25 | MATH500 | GSM8k | Olympiad | GaoKao | Minerva | AMC | Avg | |
| 512 | 32k | 29.2 | 25.2 | 84.1 | 86.7 | 45.6 | 75.5 | 26.0 | 65.7 | 54.8 | 4773 |
| | 16k | 29.0 (▼0.2) | 25.1 (▼0.1) | 84.0 (▼0.1) | 86.7 (▼0.0) | 45.5 (▼0.1) | 75.3 (▼0.2) | 26.0 (▼0.0) | 65.7 (▼0.0) | 54.7 (▼0.1) | 4441 |
| | 8k | 25.6 (▼3.6) | 23.6 (▼1.6) | 82.8 (▼1.3) | 86.7 (▼0.0) | 43.9 (▼1.7) | 74.2 (▼1.3) | 25.8 (▼0.2) | 62.4 (▼3.3) | 53.1 (▼1.7) | 3677 |
| 1024 | 32k | 30.5 | 23.9 | 84.5 | 86.3 | 47.6 | 74.2 | 28.4 | 66.5 | 55.2 | 4698 |
| | 16k | 30.5 (▼0.0) | 23.8 (▼0.1) | 84.5 (▼0.0) | 86.2 (▼0.1) | 47.4 (▼0.2) | 74.1 (▼0.1) | 27.8 (▼0.6) | 66.4 (▼0.1) | 55.1 (▼0.1) | 4409 |
| | 8k | 27.9 (▼2.6) | 22.8 (▼1.1) | 82.9 (▼1.6) | 86.2 (▼0.1) | 45.4 (▼2.2) | 73.1 (▼1.1) | 27.8 (▼0.6) | 63.0 (▼3.5) | 53.6 (▼1.6) | 3643 |
| 2048 | 32k | 30.8 | 24.6 | 84.0 | 86.1 | 46.7 | 74.3 | 27.6 | 69.2 | 55.4 | 4972 |
| | 16k | 30.8 (▼0.0) | 24.5 (▼0.1) | 83.8 (▼0.2) | 86.1 (▼0.0) | 46.5 (▼0.2) | 74.2 (▼0.1) | 27.6 (▼0.0) | 69.1 (▼0.1) | 55.3 (▼0.1) | 4518 |
| | 8k | 27.2 (▼3.6) | 23.6 (▼1.0) | 82.7 (▼1.3) | 86.1 (▼0.0) | 42.7 (▼4.0) | 73.2 (▼1.1) | 27.1 (▼0.5) | 65.6 (▼3.6) | 53.4 (▼2.0) | 3676 |
| $+\infty$ | 32k | 31.3 | 24.8 | 85.1 | 86.9 | 47.7 | 73.8 | 28.0 | 66.9 | 55.6 | 5255 |
| | 16k | 31.3 (▼0.0) | 24.7 (▼0.1) | 85.0 (▼0.1) | 86.9 (▼0.0) | 47.2 (▼0.5) | 73.8 (▼0.0) | 27.9 (▼0.1) | 66.8 (▼0.1) | 55.4 (▼0.2) | 4743 |
| | 8k | 27.5 (▼3.8) | 23.0 (▼1.8) | 82.8 (▼2.3) | 86.5 (▼0.4) | 45.4 (▼2.3) | 72.9 (▼0.9) | 27.4 (▼0.6) | 63.0 (▼3.9) | 53.6 (▼2.0) | 3938 |

## C.5. Computational Efficiency Analysis

While our approach incurs a marginal increase in computational cost compared to rule-based baselines due to the necessity of model inference, it yields a substantial improvement in data utilization. Rule-based methods frequently discard sample groups that exhibit zero reward variance, resulting in a significantly lower effective data ratio. In contrast, RLRR exploits all generated rollouts by introducing fine-grained relative rankings, ensuring maximum training efficiency per step.

Furthermore, our method demonstrates superior inference efficiency compared to the SRM. While SRMs operate under a pointwise paradigm that requires evaluating each response individually, the Ranking RM processes $n$ samples simultaneously within a single inference pass. This batch processing capability significantly reduces the overhead associated with reward calculation; **consequently, the Ranking RM exhibits no significant disparity in computational overhead compared to the SRM, and in fact, achieves a slight reduction in resource usage.** Table 9 presents an empirical estimation of the resources required for training 100 steps on A100 GPUs.

*Table 9.* Comparison of data efficiency and computational resource usage estimated over 100 training steps on A100 GPUs.

| Method | Effective Data Ratio | Training Resources (GPU Hours) |
| --- | --- | --- |
| Rule-based Verifier | 48% | 72.8 |
| Scalar Reward Model | 100% | 85.5 |
| Ranking Reward Model (Ours) | **100%** | **79.1** |

## C.6. Analysis of Ranking Reliability

We investigate the necessity of model-based relative ranking by introducing a random baseline that also incorporates correctness-aware hierarchy. In the "Random Ranking w/ Correctness Re-ranking" setting, the strict prioritization of correct samples over incorrect ones is maintained, but the relative order among responses sharing the same correctness label is

*Table 10.* Ablation study on the effectiveness of ranking mechanisms. **bold** indicates the best result.

| Method | AIME24 | AIME25 | Minerva | AMC | Avg |
|---|---|---|---|---|---|
| GRPO | 28.2 | 23.6 | 27.0 | 66.2 | 36.3 |
| Random Ranking w/ Correctness Re-ranking | 29.9 | 24.9 | 27.2 | 68.0 | 37.5 |
| **Ranking Reward Model (Ours)** | **30.8** | **24.5** | **27.6** | **69.1** | **38.0** |

## Question

Let $\triangle ABC$ be a right triangle with $\angle A = 90°$ and $BC = 38$. There exist points $K$ and $L$ inside the triangle such that $AK = AL = BK = CL = KL = 14$. The area of the quadrilateral $BKLC$ can be expressed as $n\sqrt{3}$ for some positive integer $n$. Find $n$.

## RLRR[CoT Length: 6136 tokens]

Given a right triangle $\triangle ABC$ with $\angle A = 90°$ and hypotenuse $BC = 38$. Points $K$ and $L$ inside the triangle satisfy $AK = AL = BK = CL = KL = 14$. We need to find the area of quadrilateral $BKLC$ expressed as $n\sqrt{3}$ and determine $n$.
1. **Identify the coordinates of points $K$ and $L$**:
 - Using the given conditions, we find the coordinates of $K$ and $L$ by solving the equations derived from the distances and the geometry of the triangle.
 - Points $K$ and $L$ are determined to be $K(2, 8\sqrt{3})$ and $L(13, 3\sqrt{3})$.
2. **Use the shoelace formula to find the area of quadrilateral $BKLC$**:\n - The coordinates of points $B(0, 16\sqrt{3})$, $K(2, 8\sqrt{3})$, $L(13, 3\sqrt{3})$, and $C(26, 0)$ are used.
 - Apply the shoelace formula to calculate the area:\n - Compute the sums Sum1 and Sum2 for the coordinates.
 - Calculate the absolute difference and divide by 2 to find the area.
3. **Calculate the area using the shoelace formula**:
 - Compute the sums for the shoelace formula:
 - Sum1 $= 422\sqrt{3}$
 - Sum2 $= 214\sqrt{3}$
The area is $\frac{|422\sqrt{3}-214\sqrt{3}|}{2} = 104\sqrt{3}$.
Thus, the value of $n$ is $\boxed{104}$

## GRPO[CoT Length: 7082 tokens]

Given a right triangle $\triangle ABC$ with $\angle A = 90°$ and hypotenuse $BC = 38$. Points $K$ and $L$ inside the triangle satisfy $AK = AL = BK = CL = KL = 14$. We need to find the area of quadrilateral $BKLC$.
1. **Coordinates and Equations**:
 - Assign coordinates: $A(0, 0)$, $B(b, 0)$, $C(0, c)$ with $b^2 + c^2 = 38^2 = 1444$.
 - Points $K$ and $L$ have coordinates $(x_1, y_1)$ and $(x_2, y_2)$ respectively.
 - Using the given distances, we derive equations for $x_1, y_1, x_2, y_2$ and solve for $b$ and $c$.\n\n
2. **Solving for $b$ and $c$**:
 - Through geometric reasoning and solving the equations, we find $b = 26$ and $c = 16\sqrt{3}$.
3. **Coordinates of $K$ and $L$**:
 - $K$ is at $(13, 3\sqrt{3})$ and $L$ is at $(2, 8\sqrt{3})$.
4. **Area Calculation using Shoelace Formula**:
 - Vertices of quadrilateral $BKLC$ are $B(26, 0)$, $K(13, 3\sqrt{3})$, $L(2, 8\sqrt{3})$, and $C(0, 16\sqrt{3})$.
 - Applying the shoelace formula, the area is calculated as $104\sqrt{3}$.
Thus, the area of quadrilateral $BKLC$ is $104\sqrt{3}$, so $n = \boxed{104}$

*Figure 10.* Performance on mathematical reasoning tasks, highlighting fewer reasoning tokens and clearer solution paths.

randomized. This allows us to isolate the specific contribution of the Ranking RM from the benefits provided by the correctness constraint.

As shown in Table 10, the introduction of correctness re-ranking yields an incremental improvement over the standard GRPO baseline, with the average accuracy rising from 36.3% to 37.5%. This result confirms that enforcing a hard constraint on correctness provides a stable foundation for the advantage signal. Furthermore, the Ranking RM achieves the best performance with an average accuracy of 38.0%. The gain observed over the random baseline demonstrates that the Ranking RM effectively captures nuanced quality differences within the same correctness category, providing a more reliable optimization signal than mere random ordering.

### C.7. Case Study

We present the performance on mathematical data in Figure 10. Since RLRR encourages exploring the optimal reasoning path while ensuring correctness, the number of reasoning tokens is relatively low, and the solution approach is clearer.

