# OpenReview forum: "From Absolute to Relative: Rethinking Reward Shaping in Group-Based Reinforcement Learning"
_ICML.cc/2026/Conference — ICML 2026 regular_

### Official Review · Reviewer_jcTM · 2026-02-24

**Soundness:** 2
**Presentation:** 3
**Significance:** 2
**Originality:** 2
**Overall Recommendation:** 3
**Confidence:** 3

**Summary:**

The paper introduces Reinforcement Learning with Relative Rewards (RLRR), which trains LLMs with group-based RL using relative rankings instead of absolute reward scores to avoid sparse/vanishing signals in verifiable tasks and score-instability in open-ended tasks. It proposes Hybrid Relative Reward (HRR) and Pure Relative Reward (PRR), plus advantage clipping to protect correct outputs. A listwise Ranking Reward Model and hierarchical re-ranking (correctness, then conciseness) produce reliable ranks. Experiments show consistent improvements and better best-of-N selection, and the gains transfer beyond GRPO.

**Compliance With Llm Reviewing Policy:**

Affirmed.

**Key Questions For Authors:**

1. The paper would benefit from a more thorough discussion of how performance depends on the Ranking Reward Model, including its limitations and potential failure
2. While the experimental section is extensive, it can feel overwhelming. It may help to clearly state the key questions upfront and then summarize the main takeaways (or in a short “Key Findings” paragraph).
3. It is not fully clear why answer length is incorporated into the ranking/reshaping pipeline.
4. I may have missed it, but the hyperparameter selection procedure is unclear.

**Limitations:**

The limitations are not discussed.

**Strengths And Weaknesses:**

Strengths:
1. The paper is well presented, with strong visualizations that make the method easy to follow.
2. The experiments are extensive and provide solid support for the main claims.

Weaknesses:
1. The approach may be sensitive to the quality of the Ranking Reward Model: performance could degrade if the ranking signal is noisy or biased. Moreover, training and deploying the Ranking RM introduces additional computational and engineering overhead.

---

> ### Author Rebuttal · Authors · 2026-03-29
>
> > W1: Ranking RM sensitivity & overhead
>
> We appreciate this concern. Our framework contains built-in defenses against unreliable ranking signals. First, Hierarchical Re-ranking enforces that correct responses always outrank incorrect ones, guaranteeing directional correctness regardless of RM quality. Second, Table 10 establishes a degradation lower bound: even when the Ranking RM is replaced by random ordering, RLRR still outperforms GRPO, with the full Ranking RM providing further gains. Third, as shown in Table 2 and Table 5, converting existing SRM scores (Skywork-8B, URM-8B) into relative rankings also yields consistent improvements, indicating that the core benefit stems from the relative ranking paradigm itself rather than dependence on any specific RM.
>
> To further quantify robustness under noisy rewards, we simulate conflicting feedback by flipping the correctness label with 50% probability during training:
>
> ||AIME24|AIME25|MATH500|Avg|
> |-|-|-|-|-|
> |GRPO|28.8|22.3|81.2|44.1|
> |RLRR|29.4|23.2|82.5|45.0|
>
> Even under severe label noise, RLRR consistently outperforms GRPO across all benchmarks, demonstrating strong resilience to noisy or biased reward signals.
>
> Regarding computational overhead, although the listwise input is longer than a single pointwise query, the Ranking RM processes all $n$ responses in one forward pass rather than $n$ separate evaluations. As reported in Table 9, this results in comparable or slightly lower GPU hours than the SRM baseline.
>
> > Q1: RM dependence & failure modes
>
> Thank you for the suggestion. Table 10 quantifies performance dependence: even under worst-case RM failure (random ranking), RLRR still outperforms GRPO, demonstrating bounded degradation. This robustness arises from the hierarchical design: correctness re-ranking provides the dominant signal independent of RM quality, while the RM only refines ordering within the same correctness group, bounding the impact of RM errors. The RM does inherit position bias common to listwise models, which we mitigate via correctness re-ranking. We will add a discussion in revision.
>
> > Q2: Experiment organization & key takeaways
>
> Thank you for the constructive suggestion. We will add a "Key Findings" summary in the revision:
> 1. Relative rankings consistently improve performance on both verifiable reasoning (Table 1) and open-ended generation (Table 2).
> 2. Gains do not depend on the Ranking RM; converting SRM scores into rankings already yields substantial improvements (Table 2, Table 5).
> 3. RLRR generalizes to CISPO and RLOO beyond GRPO (Table 4).
> 4. RLRR maintains effective learning signal on easy data where GRPO suffers from gradient depletion (Figure 5).
>
> > Q3: Motivation for length re-ranking
>
> Thank you for raising this point. Length re-ranking is motivated by two observations. First, among equally correct responses, a concise answer implies a more efficient reasoning path. Second, due to potential verbosity bias in reward models, longer outputs may receive higher scores regardless of quality, which could propagate into optimization without correction. We apply a coarse-grained discretization (Eq. 6) exclusively to correct responses: only length differences exceeding $\lambda$ (default 2048 tokens) alter the ranking, and incorrect responses are never promoted. Table 1 (Avg Length columns) confirms that length re-ranking reduces output length while maintaining accuracy. We will expand this motivation more clearly in the revision.
>
> > Q4: Hyperparameter selection
>
> Thank you for pointing this out. Our method involves two primary hyperparameters: $\tau$ and $\lambda$. Both are analyzed in Figure 7 and Table 8.
>
> For $\tau$, we scan over $\{0.1, 0.3, 0.5, 0.8\}$ on the 1.5B model. Figure 7(a) shows nearly identical performance at $\tau = 0.1$ and $\tau = 0.3$, with degradation only when $\tau \geq 0.5$. We select $\tau = 0.1$ as a conservative default. For $\lambda$, we scan over $\{512, 1024, 2048, +\infty\}$. Figure 7(b) shows approximately monotonic improvement with increasing $\lambda$, converging to the unconstrained case. We select $\lambda = 2048$ to balance conciseness and performance.
>
> All hyperparameters are tuned on 1.5B model and directly transferred to the 8B model and writing tasks without re-tuning, demonstrating strong transferability across scales.
>
> > Limitations
>
> We agree that this is an important addition. We will add a Limitations section covering the failure modes discussed in Question 1 above.

---

> > ### Author Rebuttal · Reviewer_jcTM · 2026-04-03
> >
> > Thank the authors for the detailed responses. I would like to keep my score unchanged. I have read the other reviewers’ comments and will keep track of the discussion between the authors and other reviewers.

---

> > > ### Author Response · Authors · 2026-04-05
> > >
> > > We sincerely thank the reviewer for the careful reading, constructive feedback, and continued engagement throughout the review process. We are especially grateful that our rebuttal has helped clarify the concerns raised.
> > >
> > > We also greatly appreciate the reviewer's positive assessment of the paper's presentation and experimental validation. The questions regarding the robustness of the Ranking Reward Model and the motivation for length re-ranking were very helpful, and they gave us a valuable opportunity to further clarify these aspects of our framework. We will also address the remaining presentation concerns in our revision, including experiment organization and hyperparameter clarity.
> > >
> > > We hope our responses have adequately addressed the concerns raised, and we would be genuinely grateful if you would be willing to reconsider your overall assessment.
> > >
> > > Thank you again for your time, thoughtful comments, and professional engagement with our work.

---

### Official Review · Reviewer_fsRh · 2026-03-12

**Soundness:** 3
**Presentation:** 2
**Significance:** 3
**Originality:** 3
**Overall Recommendation:** 4
**Confidence:** 3

**Summary:**

The authors propose a Reinforcement Learning with Relative Rewards (RLRR) framework to address the instability of original GRPO-style training, which stems from the use of point-wise reward scores. Their core idea is to utilize list-wise relative rankings instead of point-wise scores, thereby mitigating numerical instability and making the learning signal better aligned with group-based optimization.

To support this framework, they introduce a Ranking Reward Model that takes a list of responses as input and outputs their relative rankings as rewards. They also design a hierarchical re-ranking strategy to further stabilize training. Experiments on a range of tasks, including verifiable reasoning and open-ended writing, demonstrate the effectiveness of the proposed method.

**Compliance With Llm Reviewing Policy:**

Affirmed.

**Final Justification:**

My concerns about novelty are addressed. I decided to raise my score. thanks

**Key Questions For Authors:**

## Question 1: Position bias
In my view, the listwise reward model can be regarded as an enhanced preference model. However, preference models are known to suffer from position bias, meaning they may systematically favor earlier responses in the input order. Have you tested whether the predicted rankings remain consistent under permutations of the response order? If not, this would be an important analysis to include. If such bias exists, how would you mitigate it?

## Question 2: `tanh` in Eq. (3)
Could the authors provide some intuition for choosing the tanh function in Eq. (3)? Did you consider or ablate other functional forms like sigmoid?

## Question 3: Fair comparison
What is the annotation/inference cost of building the Ranking RM training data, especially when a stronger LLM is used to resolve contradictions? How could you ensure the comparison between RRM and the other reward models are fair, cuz you use some strong LLMs? Maybe the gain is just from the better data quality.

**Limitations:**

yes

**Strengths And Weaknesses:**

## Strength 1:
The paper tackles a practical and important problem, the instability of RL method. I believe the paper could contribute to the community.
## Strength 2:
The authors evaluate their method on a diverse set of benchmarks, including WritingBench and mathematical reasoning tasks and they also test it on various rl algorithms.
## Sterngth 3:
The authors also construct a new dataset, which likely required nontrivial effort and adds practical value to the work.


## Weakness 1: Cost of the RMM.
The proposed Ranking Reward Model (Ranking RM) takes a set of responses as input. However, this substantially increases the context length, roughly in proportion to the number of responses in the set. As a result, the approach may be expensive and difficult to scale to larger group sizes or longer responses.

## Weakness 2: Clarity
I found the paper somewhat difficult to follow in several places.
1. in Sec 4.1 (L154) you mentioned that *Formally, given the group of responses, we assign each response a rank ri ∈ {1, . . . , rmax} based on its relative quality, where ri = 1 denotes the best response and rmax represents the maximum rank index.* Where does the rank comes from? Does it come from the Ranking Reward Model or just change the point wise scores to the rank?
2. Could you expand your caption for all of your figures? They are confusing, especially Figure 2&3.
4. The methods presented in Tables 1 and 2 are somewhat confusing. For example, what does Skywork-8B → RLRR mean in Table 2? Similarly, what is the reward source used by RLRR(h) in Table 1? I would suggest adding a short clarifying subsection at the beginning of Sec 5 that explains each method appearing in Tables 1 and 2, including the training data, model, reward source, and how the reward signal is processed. This would make the experimental setup much easier to follow.
## Weakness 3:
Your intuition/observation that *when scalar reward models are used, large scale/range variation can destabilize relative advantage estimation.* seems not to be a new thing, which is also reported in *Pref-GRPO*. Could you elobrate the difference?

---

> ### Author Rebuttal · Authors · 2026-03-30
>
> > W1: Cost of the RRM
>
> The RRM processes all n responses in one forward pass rather than n separate calls. As reported in Tab. 9, GPU hours are comparable to or slightly lower than SRM. We provide timing comparisons under varying group sizes (Tab. B) and sequence lengths (Tab. C) in https://anonymous.4open.science/r/RLRR-results. Across all configurations, the time difference between RRM and SRM remains within an acceptable range.
>
> > W2: source of rank
>
> The rank $r_i$ in Sec 4.1 can be obtained via two pathways: 1. directly from the RRM (listwise ranking); 2. by sorting SRM pointwise scores into discrete ranks. We will clarify this in the revision.
>
> > W2: figure captions
>
> Fig. 1: (a) Sparse rewards cause intra-group variance to collapse, reducing effective training data in GRPO; (b) unbounded SRM absolute scores lead to highly variable distributions; (c) an analogy comparing gradient behavior under sparse, dense (SRM), and relative rewards.
>
> Fig. 2: The framework derives intra-group preference rankings from RM outputs with correctness and length constraints, then integrates relative reward signals via HRR or PRR depending on rule-based reward availability. A correctness consistency constraint and a clipping mechanism are applied during advantage estimation.
>
> Fig. 3: A rule-based verifier enforces that correct responses rank above incorrect ones. SRM scores determine fine-grained rankings within the same correctness category. A stronger LLM judge is invoked only when SRM rankings conflict with correctness labels. Input order is randomly shuffled to mitigate position bias.
>
> Fig. 4: Best-of-N inference scaling, where the RM selects the best response from N candidates.
>
> Fig. 5: Comparison of effective training data and performance between GRPO and RLRR across three difficulty levels.
>
> Fig. 6: Training dynamics without rule-based correctness rewards (RM signals only). (a) Rule correctness; (b) AIME2025 accuracy; (c) performance under different RM combinations.
>
> Fig. 7: Hyperparameter sensitivity analysis and effectiveness of advantage clipping.
>
>
> > W2: Tabs. 1&2
>
> We will add a notation paragraph at the beginning of Section 5. In Tab. 1, RLRR(H) uses HRR, while RLRR(P) uses PRR. In Tab. 2, the arrow variant (e.g., Skywork-8B $\to$ RLRR) converts the SRM scores into relative rankings with all other settings unchanged, isolating the benefit of ranking conversion.
>
> > W3: Comparison with Pref-GRPO
>
> While both works use relative signals, they address fundamentally different problems. Pref-GRPO targets text-to-image generation where intra-group score differences are too small, causing over-amplified advantages. RLRR addresses gradient vanishing under sparse rewards in LLM RL: when all responses are correct, rule-based verifiers yield identical scores, collapsing variance and gradients. Our HRR combines correctness signals with fine-grained relative rankings, enabling learning even in fully-correct groups. The two works are complementary rather than duplicated.
>
> > Q1: Position bias
>
> Position bias is an inherent concern in listwise models. We tested our RRM by computing the average output rank per position across multiple input permutations (ideal: 2.5). As shown in [Figure A](https://anonymous.4open.science/r/RLRR-results/position_bias.png), bias exists but remains acceptable.
>
> To mitigate this issue, during RM data construction, we randomly shuffle response orders within each group and ensure all orderings appear in balanced proportions. During RL training, we further introduce a correctness-based re-ranking stage to make the ordering more reliable.
>
> > Q2: `tanh` in Eq. (3)
>
> We chose tanh by jointly considering range and typical settings. Let Form 1 = $\tanh(r_{max}/r_i-1)$, Form 2 = $\text{sigmoid}(r_{max}/r_i-1)$, Form 3 = $2\times(\text{sigmoid}(r_{max}/r_i-1)-0.5)$. The variance of relative reward scores (ranks 1 through $n$, ignoring $\tau$) under each form is shown below. Form 2 (range [0.5,1]) has poor discriminability; only after shifting and scaling (Form 3) does variance approach Form 1. Given its simplicity and discriminability, we chose Form 1.
>
> |Form|1|2|3|
> |-|-|-|-|
> |8|0.37|0.18|0.36|
> |16|0.36|0.18|0.40|
>
> > Q3: Fair comparison
>
> The construction cost of training data is modest. Across our entire dataset, under 20% of samples conflict. We used Gemini-2.5-Flash to resolve these conflicts at a total cost below \$25, and the final training set contains only 25k samples. By comparison, our primary baseline Skywork-Reward-V2 was trained on 26 million preference pairs, partially involving human annotation. Our entire pipeline involves no human intervention, so the gains cannot be attributed to higher-quality teacher models.
>
> We reiterate that the RRM is only one means of obtaining relative rankings. As shown in Tabs. 2&5, converting existing SRM scores into relative rankings also yields consistent improvements, confirming the benefit stems from the relative ranking paradigm itself rather than data quality.

---

> > ### Author Rebuttal · Reviewer_fsRh · 2026-04-04
> >
> > Thanks for the authors’ valuable feedback. However, I would like to maintain my current score for now.
> > The distinction between your method and Perf-GRPO is still unclear to me, despite the domain, t2i or pure text. In particular, I am not sure whether the reported gains come from avoiding over-amplified advantages or from mitigating gradient vanishing under sparse rewards.
> > I have also read the other reviewer’s comments and will continue following the discussion.

---

> > > ### Author Response · Authors · 2026-04-05
> > >
> > > Thank you for the follow-up. We are grateful that our initial rebuttal resolved your concerns on computational cost, presentation clarity, and experimental fairness. The comparison with Pref-GRPO is worth clarifying carefully, and we are glad to address it directly below. We will also sharpen this positioning in our revision.
> > >
> > > In the verifiable reasoning setting, where rule-based rewards are binary, the gain comes from mitigating gradient vanishing under homogeneous rule rewards. In the open-ended writing setting, where no rule-based verifier exists and the only reward signal comes from scalar reward models, the gain comes from converting unstable, unbounded scores into bounded relative signals. These two regimes are each distinct from the setting Pref-GRPO is designed for.
> > >
> > > The key distinction lies in the specific failure mode being addressed in each case. Pref-GRPO makes a valuable observation: when pointwise reward scores within a group are very close, normalization can over-amplify these small gaps into large, destabilizing advantages, and its preference-based formulation provides an effective solution for this regime. Our two settings differ in complementary ways. In the verifiable reasoning setting, rule-based rewards frequently become completely identical within a group when all responses are correct, producing zero variance rather than small variance. There are no gaps to amplify; the group collapses to a degenerate case and GRPO loses its training signal entirely. In the open-ended writing setting, scalar reward model scores are unbounded by design. As shown in Appendix B.2, when one response receives an extreme score, it shifts the group mean and propagates bias to every other advantage estimate in the group, regardless of those responses' actual quality. This is a structural consequence of using unbounded absolute scores in group-based normalization, not a setting-specific empirical observation. The resulting failure mode points in roughly the opposite direction from what Pref-GRPO targets, and we view the two mechanisms as largely complementary.
> > >
> > > Our framework's design reflects this progression. One natural remedy for sparse rule rewards is to introduce a reward model, which densifies the signal. However, this in turn brings a second problem: pointwise reward model scores can be unstable and unbounded, leading to poorly scaled advantages, as analyzed in Appendix B. Relative rewards address both parts: they recover non-zero training signal when correctness rewards alone become uninformative within a group, and they transform unstable absolute scores into bounded relative signals for the open-ended regime.
> > >
> > > We hope the above makes clear that RLRR addresses failure modes that are qualitatively distinct from those targeted by Pref-GRPO, and that our contributions stand on independent technical grounds. We are grateful that our initial rebuttal addressed your other concerns, and we would be genuinely grateful if you would be willing to reconsider your overall assessment.

---

### Official Review · Reviewer_Qpoc · 2026-03-13

**Soundness:** 3
**Presentation:** 3
**Significance:** 3
**Originality:** 3
**Overall Recommendation:** 4
**Confidence:** 4

**Summary:**

This paper introduces RLRR, a novel framework that replaces absolute scoring with relative ranking in group-based reinforcement learning. RLRR addresses challenges like sparse supervision and score instability by incorporating intra-group rankings into the reward computation. Experimental results show that RLRR outperforms traditional methods like GRPO, achieving higher accuracy and efficiency across multiple benchmarks.

**Compliance With Llm Reviewing Policy:**

Affirmed.

**Final Justification:**

The author's rebuttal has addressed my concerns.

**Key Questions For Authors:**

N/A

**Limitations:**

The paper does not seem to include a dedicated discussion of its limitations.

**Strengths And Weaknesses:**

**Strengths:**

This approach successfully addresses key challenges such as sparse supervision and score instability. Extensive experiments demonstrate that RLRR consistently outperforms traditional methods across a range of benchmarks, including reasoning tasks and open-ended generation. The use of Ranking RM further enhances the stability and robustness of reward signals, which strengthens the overall framework.

**Weaknesses:**

While the new method brings valuable improvements to reward shaping, the connection between the theoretical analysis and real-world applications, particularly in dynamic environments, could be clearer. The experiments are limited to specific benchmarks, and it would be useful to evaluate the approach in a broader range of real-world scenarios to better assess its generalizability. Additionally, the reliance on rule-based correctness for some tasks may not fully capture the complexity of more intricate decision-making processes, and exploring more nuanced ways to handle incomplete or conflicting feedback could improve the approach further.

Additionally, the code assets linked on the homepage appear to be unavailable. There is no readme file accompanying the code (only has a readme for introducing this paper), and reviewers couldn't find references to the RLRR or the corresponding model's bash scripts mentioned in the paper (e.g., the authors claim to conduct experiments on DeepSeek-R1-Distill-Qwen-1.5B and DeepSeek-R1-Distill-LlaMA-8B). It would be helpful for the authors to provide the correct code that matches the model described in the paper.

---

> ### Author Rebuttal · Authors · 2026-03-30
>
> > W1: While the new method brings valuable improvements to reward shaping, the connection between the theoretical analysis and real-world applications, particularly in dynamic environments, could be clearer. The experiments are limited to specific benchmarks, and it would be useful to evaluate the approach in a broader range of real-world scenarios to better assess its generalizability. Additionally, the reliance on rule-based correctness for some tasks may not fully capture the complexity of more intricate decision-making processes, and exploring more nuanced ways to handle incomplete or conflicting feedback could improve the approach further.
>
> Thank you for the constructive and detailed feedback. We sincerely appreciate the reviewer's thoughtful suggestions and address each concern below.
>
> **Theory-to-practice connection.**  Appendix B directly motivates our design: Sec B.1 proves that SRM training drives reward magnitudes toward divergence, explaining the erratic reward range in Figure 1(b); Sec B.2 shows that a single outlier propagates bias through group statistics, distorting all advantages; Sec B.3 proves that RLRR bounds reward variance (PRR: $\text{Var} \leq 0.25$; HRR: bounded by $\tau$). Figure 1(b) empirically confirms that the RLRR reward range stays stable while SRM fluctuates by orders of magnitude. We agree that this correspondence could be made more explicit and will do so in the revision.
>
> **Broader evaluation.**  Our experiments already span 7 math benchmarks, 2 logic benchmarks, and WritingBench (6 domains, 100 sub-domains), across 3 model families (Deepseek, Qwen3, Qwen2.5), 4 scales (1.5B, 1.7B, 7B, 8B), and 3 RL algorithms (GRPO, CISPO, RLOO). To further strengthen generalizability, we provide an additional robustness experiment under noisy rewards. Specifically, we simulate conflicting feedback by flipping the correctness label with 50% probability during training:
>
> ||AIME24|AIME25|MATH500|Avg|
> |-|-|-|-|-|
> |GRPO|28.8|22.3|81.2|44.1|
> |RLRR|29.4|23.2|82.5|45.0|
>
> This experiment directly tests the resilience of RLRR under the "incomplete or conflicting feedback" scenario raised by the reviewer. Even with 50% label noise, RLRR consistently outperforms GRPO across all benchmarks. For future work, we plan to explore the application of RLRR in dynamic environments such as Agentic RL, where reward signals are more complex and non-stationary.
>
> **Handling conflicting feedback.** We thank the reviewer for raising this important point. RLRR incorporates two safeguards against noisy or conflicting signals: (1) Hierarchical Re-ranking enforces a hard constraint that correct responses always outrank incorrect ones, ensuring directional correctness regardless of RM quality; (2) Correctness-Aware Advantage Clipping (Eq. 5) prevents correct-but-low-ranked responses from receiving excessive penalties. Additionally, the PRR variant operates entirely without rule-based correctness, relying solely on relative rankings, and is thus directly applicable to tasks where rule verification is unavailable. As shown in Appendix C.6, even under a worst-case setting where the Ranking RM is replaced by random ordering (retaining only correctness re-ranking), RLRR still improves over GRPO, further demonstrating the robustness of our framework.
>
> > W2: Additionally, the code assets linked on the homepage appear to be unavailable. There is no readme file accompanying the code (only has a readme for introducing this paper), and reviewers couldn't find references to the RLRR or the corresponding model's bash scripts mentioned in the paper (e.g., the authors claim to conduct experiments on DeepSeek-R1-Distill-Qwen-1.5B and DeepSeek-R1-Distill-LlaMA-8B). It would be helpful for the authors to provide the correct code that matches the model described in the paper.
>
> We sincerely apologize for the confusion and thank the reviewer for bringing this to our attention. The original repository was in fact accessible and already contained the core implementation under the `./verl/` directory, which is built on VeRL. However, we fully acknowledge that the lack of training scripts and detailed usage instructions made reproduction difficult. In the updated repository at [https://anonymous.4open.science/r/RLRR-Revision](https://anonymous.4open.science/r/RLRR-Revision), we have added complete training scripts for both the Ranking RM and the RL pipeline, along with a README that describes specific usage. We hope this addresses the reviewer's concern.
>
> > Limitations
>
> We appreciate this thoughtful observation. We will incorporate feedback from all reviewers and add a dedicated Limitations section in the revision. One key limitation we plan to discuss is that the Ranking RM, as a listwise model, inherits the position bias common to listwise approaches. We mitigate this through random shuffling of input order and correctness re-ranking, and we believe further investigation of debiasing strategies is a promising direction for future work.

---

> > ### Author Rebuttal · Reviewer_Qpoc · 2026-04-04
> >
> > Thank you for your response. I have no additional questions about this submission.

---

> > > ### Author Response · Authors · 2026-04-05
> > >
> > > We sincerely thank the reviewer for the thoughtful and constructive evaluation of our work. We are especially grateful that our rebuttal helped clarify the concerns raised during review, and we sincerely appreciate the reviewer's updated score and positive acknowledgment following our rebuttal.
> > >
> > > We are glad that the clarifications on the theory-to-practice connection, the breadth of our evaluation, and the updated code repository addressed your concerns. We will incorporate these improvements in the revision to further strengthen the paper's clarity and reproducibility.
> > >
> > > Thank you again for your time, careful reading, and constructive engagement throughout the review process.

---

### Official Review · Reviewer_hH4p · 2026-03-14

**Soundness:** 3
**Presentation:** 3
**Significance:** 2
**Originality:** 3
**Overall Recommendation:** 4
**Confidence:** 4

**Summary:**

The paper proposes Reinforcement Learning with Relative Rewards, RLRR, a framework that shifts reward shaping from absolute scoring to relative ranking in group-based reinforcement learning. Through the use of group-based ranking as a dense reward signal and a complex reranking mechanism, RLRR addresses the sparse reward problem in RLVR methods. Experimental results on mathematical reasoning tasks and writing benchmarks show that RLRR outperforms baselines.

**Compliance With Llm Reviewing Policy:**

Affirmed.

**Key Questions For Authors:**

- Figure 1 illustrates the Hierarchical Re-Ranking with three main components: Correctness, Length, and Reward Model. The ablation of Hierarchical Re-ranking only considers the ablation of the first two components. Is there ablation results for Reward Models?
- Table 5 shows that RLRR w/ SRM with Qwen2.5-7B has a good performance. Does it suggest that ranking reward model is not necessary for RLRR?
- Whether the training resource consumption in Table 9 includes the time for training the Ranking RM? If not, please provide more detailed resource consumption data for training the Ranking RM.

**Limitations:**

Yes

**Strengths And Weaknesses:**

### Strengths
- The writing and presentation of the paper are clear and well-structured.
- RLRR integrates ranking signals into the reward structure, providing a more informative and dense reward signal that can effectively guide the learning process.
- RLRR is well-motivated and provides a novel perspective on reward shaping in reinforcement learning.
- The extensive validation across diverse domains, including both verifiable mathematical benchmarks and open-ended writing tasks, proves the methodology is robust and generalizes well to other group-based algorithms like CISPO and RLOO.

### Weaknesses
- The method relies heavily on hyperparameters such as λ (for length binning) and τ (for rank adjustments), which appear to be sensitive.
- The complexity of RLRR, including the design of the reranking mechanism, may increase the complexity of training and computational cost.
- Ranking RM relies on a separate LLM to generate the ranking dataset, especially when initial SRMs are unreliable, which adds a significant dependency on the quality of the "teacher" model used for data synthesis.

---

> ### Author Rebuttal · Authors · 2026-03-30
>
> > W1: Hyperparameter sensitivity.
>
> We provide detailed sensitivity analyses in Figure 7 and Table 8, showing that both hyperparameters exhibit robust behavior over wide ranges.
>
> For $\tau$, we scan over $\{0.1, 0.3, 0.5, 0.8\}$ on the 1.5B model. Figure 7(a) shows nearly identical performance at $\tau = 0.1$ and $\tau = 0.3$, with degradation only when $\tau \geq 0.5$. We select $\tau = 0.1$ as a conservative default, and advantage clipping provides additional robustness at larger values.
>
> For $\lambda$, we scan over $\{512, 1024, 2048, +\infty\}$. Figure 7(b) shows approximately monotonic improvement with increasing $\lambda$, converging to the unconstrained case. We select $\lambda = 2048$ to balance conciseness and performance.
>
> All hyperparameters are tuned on the 1.5B model and directly transferred to the 8B model and writing tasks without re-tuning, demonstrating strong transferability across scales.
>
> > W2: Training complexity and cost.
>
> Thank you for this concern. The Ranking RM processes all $n$ responses in one forward pass rather than $n$ separate evaluations, resulting in comparable or lower GPU hours than the SRM baseline (Table 9). The Hierarchical Re-ranking mechanism incurs negligible cost as it only involves sorting over pre-computed values. RLRR significantly increases data utilization at a modest additional cost, yielding better training efficiency overall.
>
> > W3: Dependency on teacher model quality.
>
> We acknowledge the dependency, but multiple safeguards limit its impact. 1. In data construction, the rule-based correctness verifier provides a reliable signal independent of the SRM; correctness strictly overrides SRM scores, and contradictory SRM signals are discarded. 2. During RL training, Hierarchical Re-ranking enforces the same correctness-first principle in advantage estimation. Table 10 confirms this robustness: even replacing SRM rankings with random permutations, correctness re-ranking alone improves over GRPO. We stress that RLRR's core contribution is shifting from absolute scores to relative rankings to improve data utilization, not relying on a stronger reward model. Tables 2 and 5 show RLRR w/ SRM  already outperforms absolute-score baselines.
>
> > Q1: Ablation of the RM component?
>
> The ablation of the Reward Model component is already covered in our existing experiments. In Table 10, the "Random Ranking w/ Correctness Re-ranking" baseline retains correctness and length hierarchies but replaces the Ranking RM with random ordering, directly ablating the RM component. Results show that correctness and length re-ranking alone already improve over GRPO, and the Ranking RM further refines the ranking by capturing fine-grained quality differences within the same correctness category. Table 5 provides a complementary view: RLRR w/ SRM replaces the Ranking RM with SRM-derived rankings and achieves comparable performance, confirming that the Ranking RM provides consistent but not dominant gains. We further extend Table 3 with an explicit RM ablation in [Table A](https://anonymous.4open.science/r/RLRR-results/README.md), confirming that the RM provides moderate but consistent gains.
>
> > Q2: Is the Ranking RM unnecessary given RLRR w/ SRM's strong performance?
>
> The strong performance of RLRR w/ SRM actually reinforces our core thesis: the primary improvement comes from shifting absolute scores to relative rankings, not from any specific reward model. RLRR is designed to be ranking-source agnostic, and RLRR w/ SRM shows that converting pointwise scores into ordinal ranks yields meaningful gains. That said, the Ranking RM offers advantages when available: (1) listwise comparison allows the model to jointly consider all responses in a group, providing richer context than SRM's independent pointwise scoring; (2) single-pass inference over all responses makes it slightly more efficient than $n$ separate SRM evaluations; (3) substantially stronger performance on open-ended tasks without rule-based verifiers. The Ranking RM is therefore not necessary but preferred when resources permit, reflecting the flexibility of our framework.
>
> > Q3: Does Table 9 include RM training cost?
>
> Table 9 compares computational costs during the RL training phase (100 steps on A100 GPUs) and does not include RM pre-training, as this would conflate two separate stages. As for the Ranking RM's training cost: we fine-tune from Qwen2.5-7B-Instruct-1M with only 25k samples for 3 epochs on 16 A100 GPUs (~20 hours), a one-time investment amortized across all subsequent RL runs. For comparison, our SRM baseline Skywork-Reward-V2 was trained on 26M preference pairs on 64 H800 GPUs [1], making the Ranking RM's cost orders of magnitude lower.
>
> [1] Liu et al. "Skywork-reward-v2: Scaling preference data curation via human-ai synergy." arXiv:2507.01352 (2025).

---

### Decision · Program_Chairs · 2026-04-30

**Decision:**

Accept (regular)

**Comment:**

This paper proposes Reinforcement Learning with Relative Rewards (RLRR), a framework designed to improve group-based reinforcement learning (like GRPO) by shifting from absolute numerical rewards to relative rankings. The authors introduce a Ranking Reward Model (RRM) and a hierarchical re-ranking strategy (prioritizing correctness, then conciseness, then quality) to provide denser and more stable training signals. The method is evaluated across a wide array of benchmarks, including mathematical reasoning and open-ended writing tasks, showing consistent improvements over absolute-reward baselines. During the rebuttal phase, the authors successfully addressed several key technical concerns including efficiency of RRM, robustness against label noise, and the novelty of the method. The paper presents a technically sound and well-validated approach to a high-interest topic in LLM alignment. Therefore, we recommend weak accept.